# Chemical engineering of therapeutic siRNAs for allele-specific gene silencing in Huntington's disease models

Faith Conroy[1,4], Rachael Miller [1,4], Julia F. Alterman[2], Matthew R. Hassler[2], Dimas Echeverria[2], Bruno M. D. C. Godinho [2], Emily G. Knox[2], Ellen Sapp[3], Jaquelyn Sousa[2], Ken Yamada[2], Farah Mahmood[3], Adel Boudi[3], Kimberly Kegel-Gleason[3], Marian DiFiglia[3], Neil Aronin[1,2], Anastasia Khvorova [2] ✉ & Edith L. Pfister [1] ✉

Small interfering RNAs are a new class of drugs, exhibiting sequence-driven, potent, and sustained silencing of gene expression in vivo. We recently demonstrated that siRNA chemical architectures can be optimized to provide efficient delivery to the CNS, enabling development of CNS-targeted therapeutics. Many genetically-defined neurodegenerative disorders are dominant, favoring selective silencing of the mutant allele. In some cases, successfully targeting the mutant allele requires targeting single nucleotide polymorphism (SNP) heterozygosities. Here, we use Huntington's disease (HD) as a model. The optimized compound exhibits selective silencing of mutant huntingtin protein in patient-derived cells and throughout the HD mouse brain, demonstrating SNP-based allele-specific RNAi silencing of gene expression in vivo in the CNS. Targeting a disease-causing allele using RNAi-based therapies could be helpful in a range of dominant CNS disorders where maintaining wild-type expression is essential.

Oligonucleotide therapeutics is a novel class of drugs advancing the treatment of a wide range of diseases[1,2]. The clinical success of these compounds is predicated on an optimized chemical scaffold supporting delivery to the tissue of interest; the sequence can then be modified to target other genes of interest[3,4]. One class of oligonucleotide therapeutics is small interfering RNAs (siRNAs). Upon loading into the RNA-induced silencing complex (RISC), siRNAs cleave complementary mRNAs leading to degradation of the disease-causing mRNA and preventing protein expression[5–8].

Stabilization of the siRNA is critical for sustained efficacy. A single injection of a multi-valent, GalNac-conjugated, fully-modified siRNA safely supports target silencing in the liver for up to 6–12 months[1,9–11]. Endosomal entrapment of oligonucleotides creates an intracellular depot and provides a continuous supply of siRNA for RISC loading[12],

explaining the long duration of effect. A combination of sugar, backbone, and terminal phosphate modifications[13–18] stabilizes the compound in the endosomal compartment and is required to ensure durable gene silencing in vivo.

Recently, we developed an optimized siRNA chemical architecture that achieves broad distribution and long-lasting silencing in the brains of rodents and non-human primates. The divalent, chemically modified (Di-siRNA) exhibited delayed cerebrospinal fluid (CSF) clearance and efficient neuronal internalization[19]. Furthermore, a single intracerebroventricular (ICV) injection of a modified siRNA (Di-siRNA) targeting the *HTT* mRNA produced widespread silencing of total *HTT* for up to six months.

Huntington's disease (HD) is an autosomal dominant neurodegenerative disorder caused by the expansion of the CAG repeat region

[1]Department of Medicine, UMass Chan Medical School, Worcester, MA 01605, USA. [2]RNA Therapeutics Institute, UMass Chan Medical School, Worcester, MA 01605, USA. [3]Department of Neurology, Massachusetts General Hospital, Harvard Medical School, Boston, MA 02114, USA. [4]These authors contributed equally: Faith Conroy, Rachael Miller. ✉e-mail: anastasia.khvorova@umassmed.edu; edith.pfister@umassmed.edu

of the Huntingtin (*HTT*) gene. Reducing huntingtin protein *(htt)* expression is a therapeutic approach[20,21] that has shown promise in multiple disease models[22–25]. Recent evidence indicates that wild-type huntingtin might be important for normal neuronal function[26,27], therefore, selective silencing of the mutant isoform might be advantageous. Moreover, a large phase III clinical trial evaluating the efficacy of a non-selective antisense oligonucleotide (ASO) targeting HTT was recently halted prematurely due to negative effects[28,29]. ASO-HTT treated patients showed an increase in ventricular volume, increased expression of neurofilament light chain (NfL), and a decrease in functional performance readouts. While the negative outcome is likely related to non-specific inflammatory effects of ASOs, the potential impacts of non-selective reduction of HTT cannot be ruled out. Like HD, many genetic disorders are dominant[30], and silencing of the mutant, but not the wild type allele, may be preferable[26,31–33]. One strategy for allelic discrimination targets SNP heterozygosities[34,35] present in normal and disease-causing genes[34,35]. While theoretically ideal, SNP-based allele-specific silencing using chemically modified siRNAs has never been demonstrated in brain.

Multiple reports describe the in vitro development and validation of unmodified siRNAs that effectively discriminate between mutant and wild-type alleles based on a single nucleotide[34–41]. However, the degree of discrimination achieved was 10-fold or less[42], which may not translate into effective discrimination in vivo where siRNA accumulation varies regionally and temporally[19]. In addition, most in vitro studies were performed using non-modified or partially modified siRNAs, which are unstable and do not distribute widely in vivo. Full chemical stabilization is essential for in vivo efficacy[35] but can significantly affect guide strand thermodynamics[43]. Thus, the chemical modification of siRNAs changes their target recognition and discrimination properties, altering their SNP targeting profile.

Using a fully chemically modified siRNA scaffold, we performed a systematic series of screens targeting two SNPs, rs362307 and rs362273, that are frequently heterozygous in HD patients[34,35]. For rs362273, we identified one siRNA exhibiting greater than 50-fold discrimination and confirmed the selectivity of this siRNA in human neurons differentiated from HD patient iPSCs. In the CNS-active (DisiRNA) conformation, the optimized siRNA showed widespread distribution in a transgenic mouse model of HD (BACHD)[44] and maintained allele-specific silencing in all brain regions, independent of dose or level of siRNA accumulation. These results demonstrate RNAi-based allele-specific silencing using fully modified siRNA in vivo. The approach—demonstrated for HD—can be applied to numerous other genetic disorders where allele-specific mutant gene silencing is necessary.

## Results

Multiple SNPs in the *htt* gene are frequently heterozygous in patients[34,35,45]. Here, we focus on two SNPs in the *htt* gene (Supplementary Fig. 1): rs362273 in exon 57, which is heterozygous in 35% of HD patients, and rs362307 in exon 67, which is heterozygous in 48% of HD patients[34,35]. One of the challenges for in vivo evaluation of SNP-selective siRNAs is access to in vivo models. For in vivo evaluation, we chose SNP rs362273, which is present in BACHD mice. The surrounding region is conserved in the mouse *htt* gene creating a model where the mutant (human, expanded CAG repeat) and the wild-type (mouse) HTT differ by a single nucleotide at the SNP site. Because the mutant allele has a long CAG repeat, the mutant and wild-type HTT proteins can be separated by western blot and evaluated simultaneously for selective silencing.

### Primary screen generates siRNAs with moderate SNP-based discrimination

We designed a panel of twelve chemically stabilized siRNAs, overlapping SNP rs362273 of the huntingtin mRNA (Fig. 1). Supplementary

Table 1 shows the sequences and chemical modification patterns of all compounds used in the study; sequences without modifications are provided in Supplementary Table 2. Alternating 2′-O-methyl (2′-OMe) and 2′-deoxy-2′-fluoro (2′-F) replaced the riboses, and terminal backbones were modified with phosphorothioates (PS); the sense strand was conjugated to cholesterol (Fig. 1a). We refer to these siRNAs as fully chemically modified. Cholesterol conjugated, chemically stabilized siRNAs are efficiently internalized in all cell types through endocytosis and lysosomal entrapment[13,46]. The cholesterol-driven self-delivery mechanism is analogous to the mechanism of in vivo uptake and requires extensive stabilization. Therefore, the same chemical scaffold, with the same pattern of 2′-F, 2′-OMe, and PS modifications, but without cholesterol, can be re-synthesized in the divalent, CNS-active configuration for a seamless transition between in vitro and in vivo experimental systems[19].

For the evaluation of on-target and non-target efficacy, we used two luciferase-based reporter plasmids containing a 40-nucleotide portion of the huntingtin mRNA surrounding SNP site rs362273[35]. Cells were transfected with either the 2273-A or 2273-G reporter plasmid. The next day, the cholesterol-conjugated siRNA was added. Figure 1b shows the results of the primary screen. The silencing of target and non-target reporters varied significantly depending on siRNA sequence and position of the SNP heterozygosity. For example, the siRNA with the SNP in position 3 showed efficacy but little discrimination, while those with the SNP in positions 12 and 13 showed limited efficacy.

For compounds with the highest on-target efficacy and discrimination, we evaluated the efficiency of target and non-target silencing at multiple concentrations and calculated IC50 values (Fig. 1c). Two siRNAs, SNP4 (SNP in position 4) and SNP6 (SNP in position 6), exhibited potent target gene silencing (IC50 ~38 and 24 nM correspondingly) and demonstrated ~20-fold (SNP4) and ~7-fold (SNP6) discrimination against the non-target reporter based on a single mismatch. We proceeded to optimize these two candidates.

### The SNP targeting siRNAs is two to four-fold less potent than the most active pan-targeting siRNAs in the native genomic context

Extensive screening across the whole gene can identify highly potent pan-targeting siRNAs but the sequence space available for identifying SNP targeting compounds is limited. By screening hundreds of siRNAs, we have previously identified a highly active non-selective siRNA[HTT] targeting HTT[13]. HeLa cells express endogenous HTT and have the A isoform at the rs362273 SNP site. Therefore, we can compare the potencies of SNP- and non-selective siRNAs against the endogenous target. We evaluated the efficacy of SNP4, SNP6, and siRNA[HTT] in HeLa cells at seven concentrations (Supplementary Fig. 2). In passive uptake, the corresponding IC50 values for SNP4, SNP6, and siRNA[HTT] were 302, 149, 73 nM, indicating that limiting the targeted sequence space by targeting a SNP resulted in at least a two-fold reduction in potency.

### Introduction of a secondary mismatch enhances SNP-based discrimination

The level of siRNA accumulation in vivo varies significantly between different brain regions and over time. Therefore, a single mismatch at the optimal position (SNP6) likely provides insufficient discrimination to support allele-specific silencing in vivo. In the context of a fully modified guide strand, a single mismatch introduces only a small thermodynamic disturbance, and indeed in certain positions, a single mismatch between the siRNA and the target mRNA can enhance efficacy[47]. We measured the impact of the mismatch in position six on the Tm of the guide strand/RNA substrate (Supplementary Fig. 3). In the context of the full-length guide, the impact of the mismatch on stability was 4 °C (85.4–81.4 °C), which is not biologically significant. In the context of a 13-mer substrate (comprising the RISC core interactions), the impact of the mismatch was greater (5.1–68.5 °C to 73.6 °C).

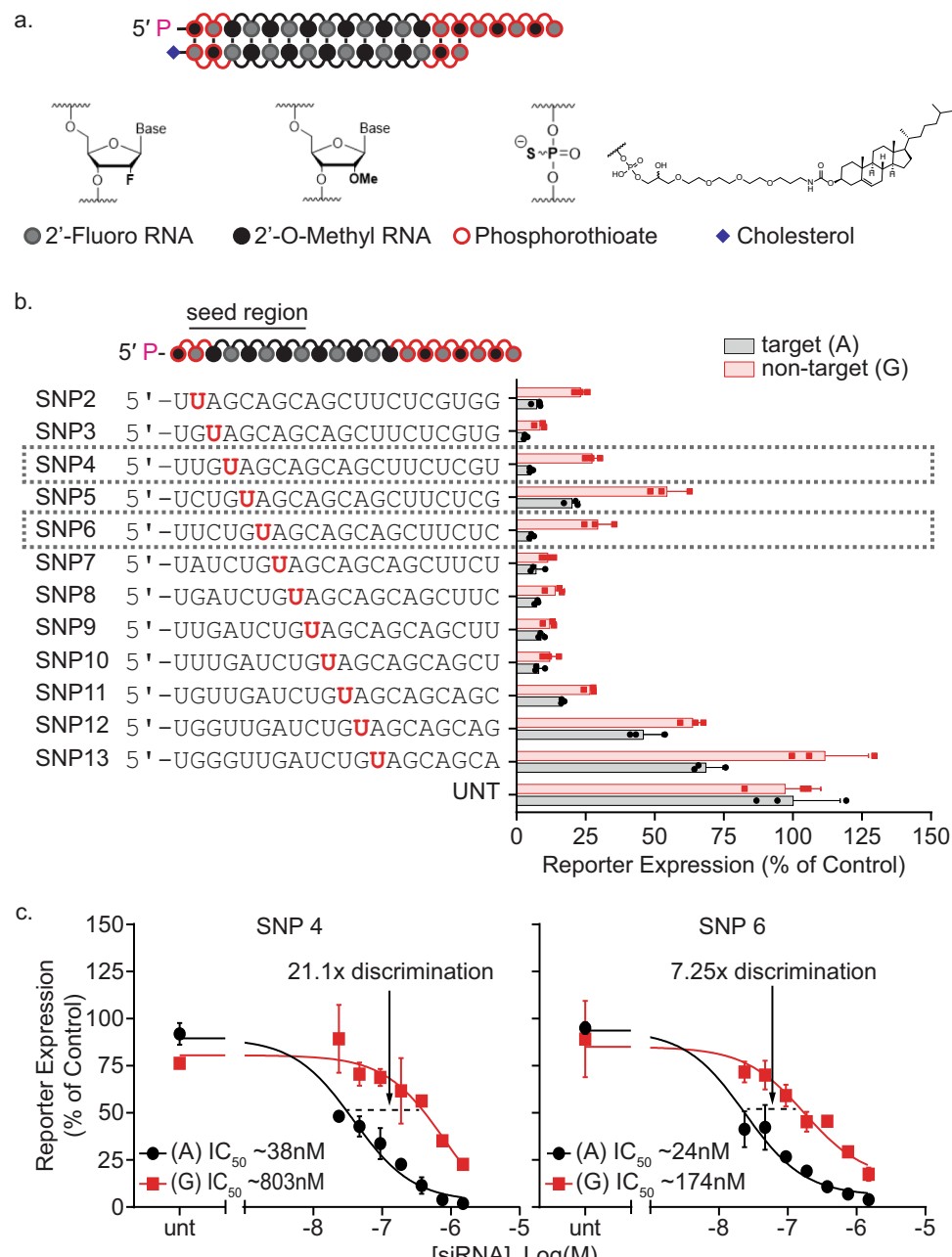

**Fig. 1 | The primary screen for optimal siRNA sequence yields potent compounds with moderate discriminating power. a** siRNA structure and chemical modification pattern used in screening. **b** The primary screen identifies the most favorable positions of the SNP enabling single nucleotide discrimination for targeting of SNP site rs362273 (highlighted in red). Compounds were tested using a dual-luciferase reporter assay system in HeLa cells. The (psiCheck) reporter plasmids contain a 40 mer region of huntingtin, including the target (A) SNP (black), and non-target (G) isoform (red). Cells were treated for 72 h at 1.5 μM of siRNA. A panel of siRNA sequences were synthesized in a cholesterol-conjugated scaffold with phosphorothioate and alternating 2′-F and 2′-OMe backbone modifications. By walking the siRNA sequence around SNP site rs362273, we find multiple compounds with varying degrees of efficacy and discrimination; $n = 3$ wells/treatment. **c** 7-point dose-response shows that siRNAs with the SNP site in positions 4 (SNP4) and 6 (SNP6) generate 20-fold and 7-fold allelic discrimination, respectively, with a high degree of efficacy; $n = 2$ wells/treatment. All data are presented as mean ± standard deviation. Source data are provided as a source data file.

However, in both cases, the affinity of the modified guide strand for the target is high. The likely explanation for the observed discrimination is a disruption of the local architecture of the siRNA-target duplex, which interferes with the ability of RISC to form the active conformation and disrupts scanning and recognition of the target by the seed region (positions 2–8)[48–50].

Introducing a secondary mismatch[35] can enhance SNP-based discrimination by modulating guide strand affinity for the target. In the context of a SNP heterozygosity, siRNAs with a secondary mismatch (Fig. 2a) have a single mismatch to the target isoform and two mismatches to the non-target isoform. The second mismatch can significantly reduce the affinity of the guide strand for the double-mismatched non-targeted isoform, resulting in a loss of silencing[35]. Using the two best siRNAs from the primary screen, SNP6 (Fig. 2b) and SNP4 (Supplementary Fig. 4), we performed a secondary mismatch screen. Introducing an intentional mismatch to the SNP6 siRNA at position 11 enhances the level of target/non-target discrimination to more than 50-fold (Fig. 2b).

Ago2 uses the seed region for primary target recognition[48]. Additional complementarity throughout the rest of the guide stand

can compensate for an imperfect seed. A secondary mismatch outside of the seed region minimally impacts target silencing but reduces the silencing of the non-targeted mRNA isoform (Fig. 2a). To quantify the differences in target and non-target recognition, we evaluated compounds with secondary mismatches in positions 11, 14, or 16 in a 7-point dose-response (Fig. 2b). None of these secondary mismatches

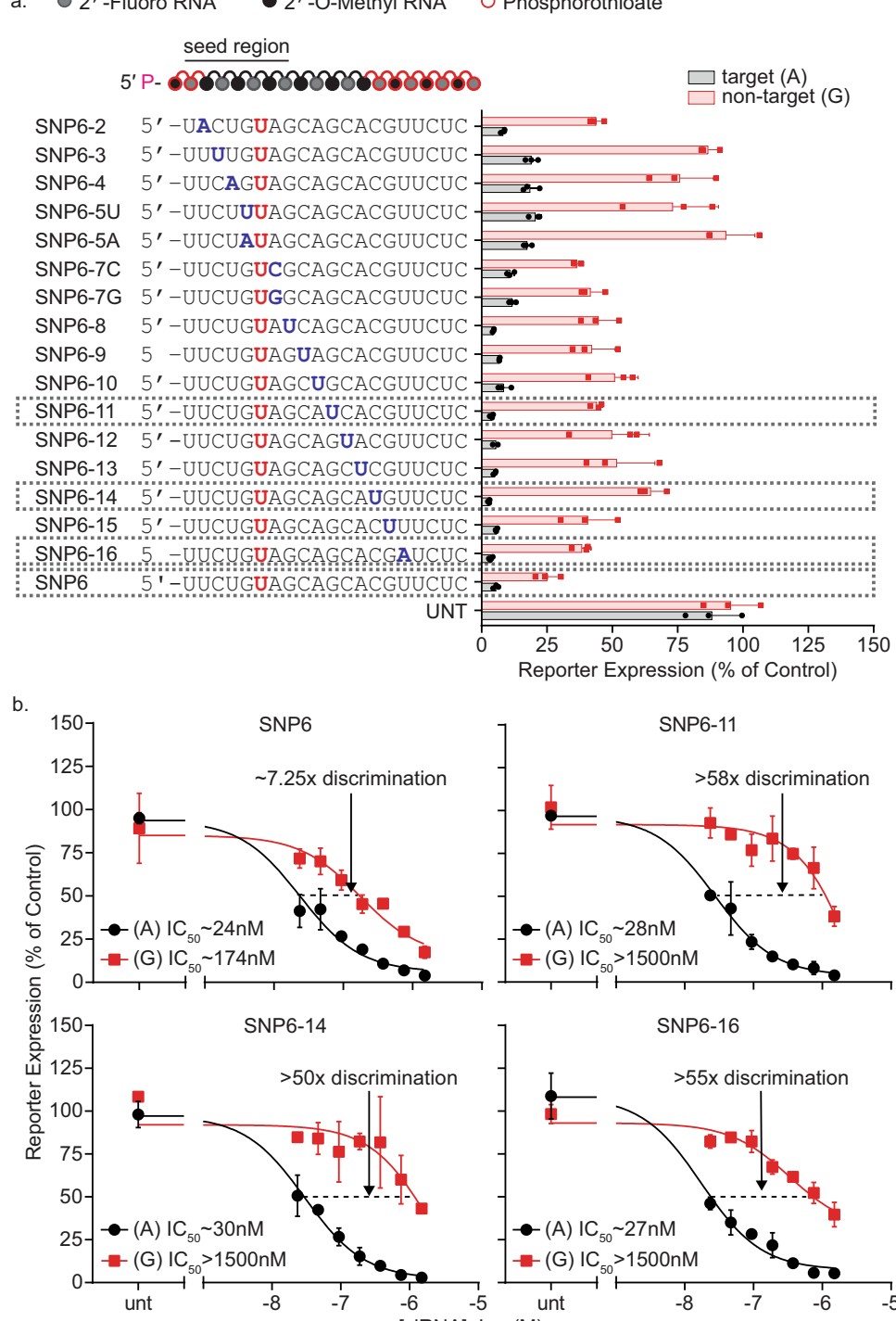

**Fig. 2 | A systematic screen of siRNAs targeting a heterozygosity at site rs362273 yields multiple compounds in which a secondary mismatch improves allelic discrimination. a** In certain positions, adding an intentional mismatch to a SNP-targeting siRNA improves discrimination without impairing target silencing. Compounds were tested using a dual-luciferase reporter assay system in HeLa cells. The (psiCheck) reporter plasmids contain a 40 mer region of huntingtin, including the target (A) SNP isoform, and non-target (G) SNP isoform. Cells were treated for 72 h at 1.5 μM of siRNA. An siRNA selected from the primary screen (SNP6) (Fig. 1) was used as a template sequence and a secondary mismatch (highlighted in blue) was introduced. When a mismatch is added outside the seed region, siRNAs show a substantial increase in discriminating power without decreasing target silencing; *n* = 3 wells/treatment. **b** 7-point dose-response of selected compounds shows that the addition of a U:C mismatch in position 11 increases discrimination from approximately 7-fold to greater than 50-fold. Changes in efficacy and discrimination vary based on the position of the secondary mismatch; *n* = 2 wells/treatment. All data are presented as mean ± standard deviation. Source data are provided as a source data file.

affected on-target silencing (IC50s ~20–30 nM). The impact on non-target silencing varied, increasing discrimination to greater than ~50-fold (Fig. 2b). The position of the best secondary mismatch is sequence-dependent; a secondary mismatch in position 7 improves discrimination when the primary mismatch was at position 4 (Supplementary Fig. 4b), and a position 11 secondary mismatch was better when the primary mismatch was at position 6 (Fig. 2b). Therefore, the exact combination of primary and secondary mismatches needs to be optimized individually for each sequence. For further optimization, we selected compound SNP6-11, which carries two mismatches and showed good on-target activity (IC50 ~28 nM), with more than 58× discrimination against the non-targeted isoform (IC50 > 1500 nM).

### Chemical modifications at the mismatch positions affect potency and discrimination
While both 2′-F and 2′-OMe stabilize the siRNA, the impact of 2′-OMe modification on stability is significantly more pronounced[51]. Thus, increases in 2′-OMe content improve siRNA stability and enhance duration of effect in vivo[16,52], but extensive 2′-OMe modifications can position and sequence-specifically decrease activity[53]. Therefore, most siRNAs designed for in vivo and clinical applications use sequence-optimized patterns of 2′-OMe and 2′-F modifications[4]. Successful development of SNP targeting siRNA requires an understanding of how the chemical modification pattern around the mismatch affects efficacy and discrimination. 2′-OMe and 2′-F affect guide strand/target interactions differently. 2′-F strengthens target interaction substantially more than does 2′-OMe[54,55]. An increase in 2′-F content, particularly around the mismatch site, is expected to increase local affinity of the siRNA for its target, reducing the impact of a mismatch. Meanwhile, the bulkier 2′-OMe might increase discrimination by reducing the silencing of the non-targeted isoform.

A panel of compounds with different combinations of 2′-OMe and 2′-F modifications near the primary (position 6) and secondary (position 11) mismatch positions were synthesized and tested. Changes in the chemical modification pattern around the mismatch sites affected potency and discrimination (Fig. 3a). Figure 3b–e shows the dose responses for the most informative configurations. Placing 2′-F modifications flanking the primary site (SNP fm6-11, Fig. 3c) maintains on-target potency (IC50 ~17 nM vs 18 nM, Fig. 3b, c) but eliminates discrimination. Increased 2′-OMe content around the secondary mismatch (SNP 6-m11) (Fig. 3d) reduces on-target silencing (IC50 ~58 nM). Combining 2′-F modifications around the primary SNP site with a methyl rich secondary mismatch region (SNP f6-m11, Fig. 3e) restores on-target activity (IC50 ~22 nM) and maintains discrimination. These data suggest that both structural and chemical contexts near the mismatch affect siRNA activity and selectivity. Therefore, SNP selectivity must be optimized in the same chemical context intended for in vivo use.

### The optimal SNP-discrimination scaffold is sequence-specific
Having identified a highly selective siRNA targeting HD SNP rs362273, we tested whether this approach could be applied to other sequences. We repeated the same workflow, this time targeting HD SNP rs362307, which is heterozygous in 48% of HD patients[35]. This SNP is not present in current HD mouse models[56], precluding in vivo evaluation.

The primary screen (Supplementary Fig. 5a) identified only one siRNA with acceptable activity (IC50 target ~35 nM) and discrimination (IC50 for non-target ~438 nM) (SNP3, Supplementary Fig. 5b). A secondary mismatch screen demonstrated that both the position and the identity of the mismatch affected potency and selectivity (Supplementary Fig. 6a). Introduction of a secondary mismatch at position 8 increased selectivity (Supplementary Fig. 6b) whereas a secondary mismatch at position 5 increased selectivity, but also decreased on target potency (Supplementary Fig. 6c). In addition to the position, the type of mismatch was

important. For example, the SNP3-7C (Supplementary Fig. 6d) compound had little activity toward the non-targeted allele (IC50 > 1500 nM) but lost up to 5-fold in target silencing efficacy (IC50 ~400 nM) relative to the single mismatch SNP3 compound. SNP3-7G (Supplementary Fig. 6e) shares the same mismatch positions as SNP3-7C but differs in sequence at the secondary mismatch position. SNP3-7G showed the same efficacy against the targeted allele as SNP3-7C but did not discriminate between alleles. Consistent with the idea that the wobble base-pair is less structurally disruptive[42], the C-U mismatch discriminates better than the G-U mismatch[57] (SNP3-7C vs. SNP3-7G, Supplementary Fig. 6d, e). The optimal scaffold SNP3-5G (Supplementary Fig. 6c) exhibited effective target silencing (IC50 ~92 nM) and efficient discrimination (non-targeted IC50 > 1500 nM). These data suggest that both the chemical and sequence context of the mismatches influence efficacy and discrimination, requiring individual optimization.

### Allele-selective lowering of mutant HTT protein in human neural stem cells and in stem cells differentiated into neurons
To determine if SNP6-11 could discriminate between wild-type and mutant huntingtin in a native human context, we treated human neural stem cells (NSCs) with SNP6-11 siRNA. The NSCs were created from induced pluripotent stem cells (iPSCs) with 109 CAG repeats (Fig. 4a). We performed DNA sequencing and determined that these cells were heterozygous at rs362273. There are three major huntingtin haplogroups, A, B, and C[58]. A majority of HD chromosomes belong to the A haplogroup, representing targetable SNPs associated with HD. In contrast, the C haplogroup is overrepresented on normal chromosomes[58]. Evidence from other SNP heterozygosities suggests that the parental CAG109 cell line belonged to the A or B haplogroups[59–61]. Both A and B haplogroups have an A on the mutant isoform at the rs362273 SNP site; since we had established heterozygosity, we reasoned that the non-HD chromosome would belong to haplotype C and have a G at the SNP site.

After treatment with siRNA, we measured the levels of wild-type and mutant HTT protein by SDS-PAGE and western blot. The non-selective siRNA[HTT] lowered both wild-type and mutant HTT protein (Fig. 4b, c). In contrast, when compared to a non-targeting control (NTC) at 3.0 μM, SNP6-11 produced a dose-responsive lowering of mutant HTT at 1.5, 2.0, and 3.0 μM (Fig. 4b) with no corresponding reduction of wild-type HTT (Fig. 4c). This data indicates that based on a single nucleotide difference and in the presence of two native human *htt* alleles, the SNP 6–11 siRNA selectively lowers mutant HTT (A) without affecting the wildtype protein (G).

Next, we asked if SNP6-11 could reduce mutant HTT protein selectively in a disease-relevant cell type. HD109 iPSCs were differentiated to cortical neurons. Western blot analysis showed that the neuronal marker βIII-tubulin was present in cell lysates and immunofluorescent labeling of parallel wells revealed a high percentage of βIII-tubulin positive cells, confirming neuronal enrichment (Supplementary Fig. 7a, b). siRNAs were added for 5 days continuously, starting on differentiation day 43, as described in Methods. Compared to NTC at 3.0 μM, SNP6-11 at 2.0 and 3.0 μM significantly lowered mutant HTT protein by 35 and 38%, respectively (Fig. 5a). In contrast, levels of wild-type HTT protein were unchanged (Fig. 5b). At 3.0 μM, non-selective siRNA[HTT] lowered both mutant and wild-type HTT protein by 80 and 78% (Fig. 5a, b). Like in HeLa cells (Supplementary Fig. 2), the SNP 6–11 siRNA shows efficient discrimination between the two alleles but lower efficacy when compared to the non-selective siRNA[HTT].

### SNP targeting siRNA compounds selectively silence mutant huntingtin in the mouse brain
The next step was to evaluate whether the SNP6-11 compound, targeting rs362273, can discriminate between mutant and wild-

 

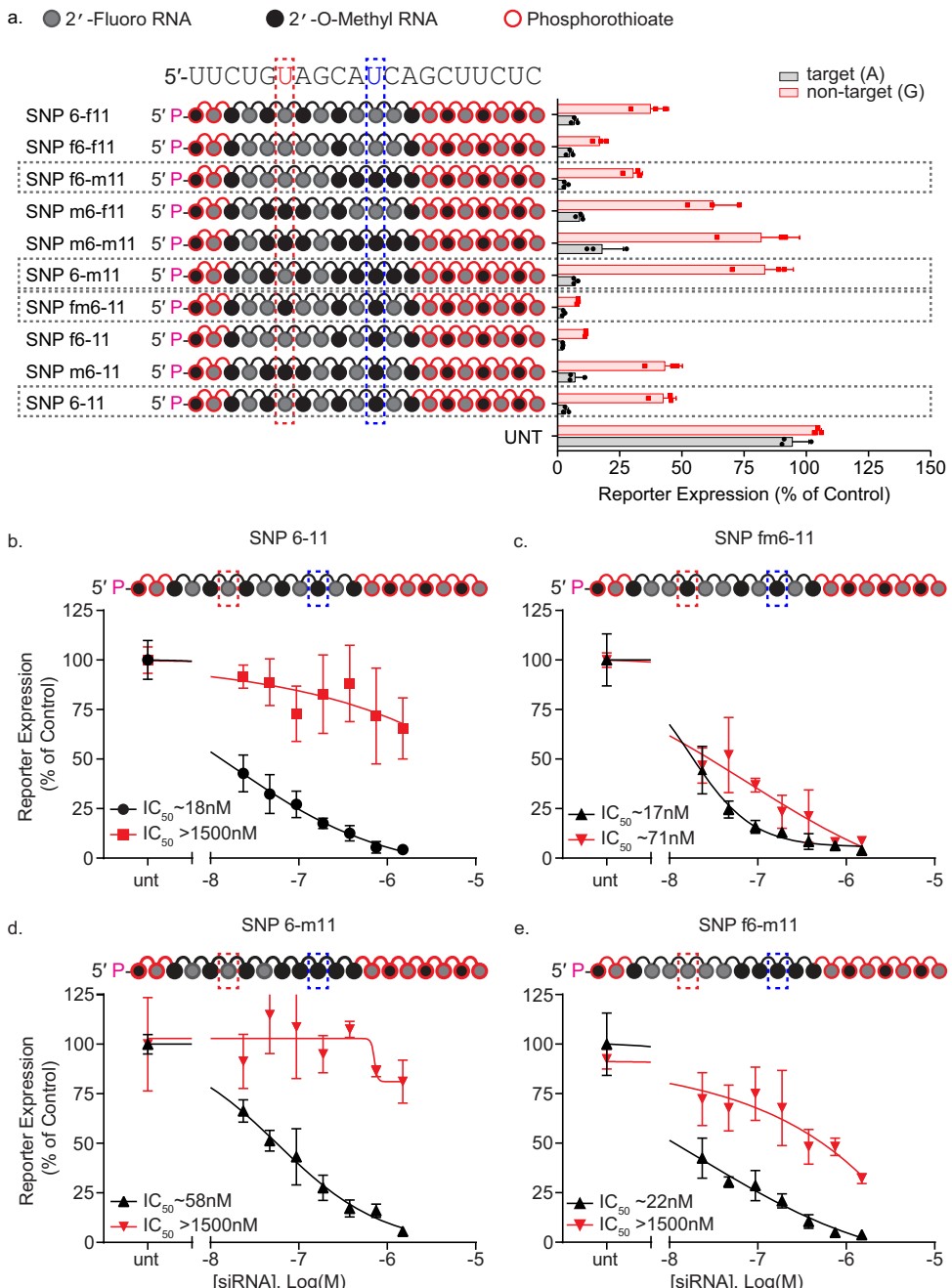

**Fig. 3 | Chemical modification pattern affects allele specificity and efficacy.**
**a** Combinations of 2′-F and 2′-OMe enrichment around critical positions 6 and 11 affect siRNA activity. Compounds were tested using a dual-luciferase reporter assay system in HeLa cells. The (psiCheck) reporter plasmids contain a 40 mer region of huntingtin, including the target (A) SNP isoform, and non-target (G) SNP isoform. Cells were treated for 72 h with 1.5 μM siRNA. **b–e** 7-point dose-responses of alternative modification patterns show that backbone modification pattern impacts efficacy and discrimination. Compared to the original alternating modification pattern (**b**), a heavily-fluorinated SNP region shows an increase in target efficacy, but a complete loss of discrimination between alleles (**c**). A heavily methylated mismatch region increases discrimination but decreases target efficacy (**d**). Combining a fluorinated SNP region with a methylated mismatch region improves efficacy but does not increase discrimination above the original alternating modification pattern (**e**); *n* = 3 biological replicates for all experiments. All data are presented as mean ± standard deviation. Source data are provided as a source data file.

type huntingtin in vivo. The BACHD mouse model expresses wild-type mouse huntingtin and overexpresses a human HD transgene[44]. The sequence overlapping rs362273 is conserved between human and mouse genes, with the mouse *htt* having an A, and the transgene a G at the SNP position (Fig. 6a). Therefore, we can evaluate selective SNP-based silencing in this model. For in vivo delivery, the SNP6-11 compound was synthesized in a divalent siRNA configuration (Fig. 6b), which supports robust silencing in the mouse and NHP brain[19]. Supplementary Tables 1

and 2 show the sequence and chemical modification patterns used.

We administered 225 μg (10 nmol, 5 nmol/ventricle) of divalent SNP6-11 into the lateral ventricles of BACHD mice. PBS and a chemically matched divalent NTC siRNA were used as negative controls. One-month post-injection, the levels of mutant human and wild-type mouse huntingtin protein were measured in the cortex, striatum, hippocampus, and thalamus. There were no significant differences between PBS and NTC, indicating that the

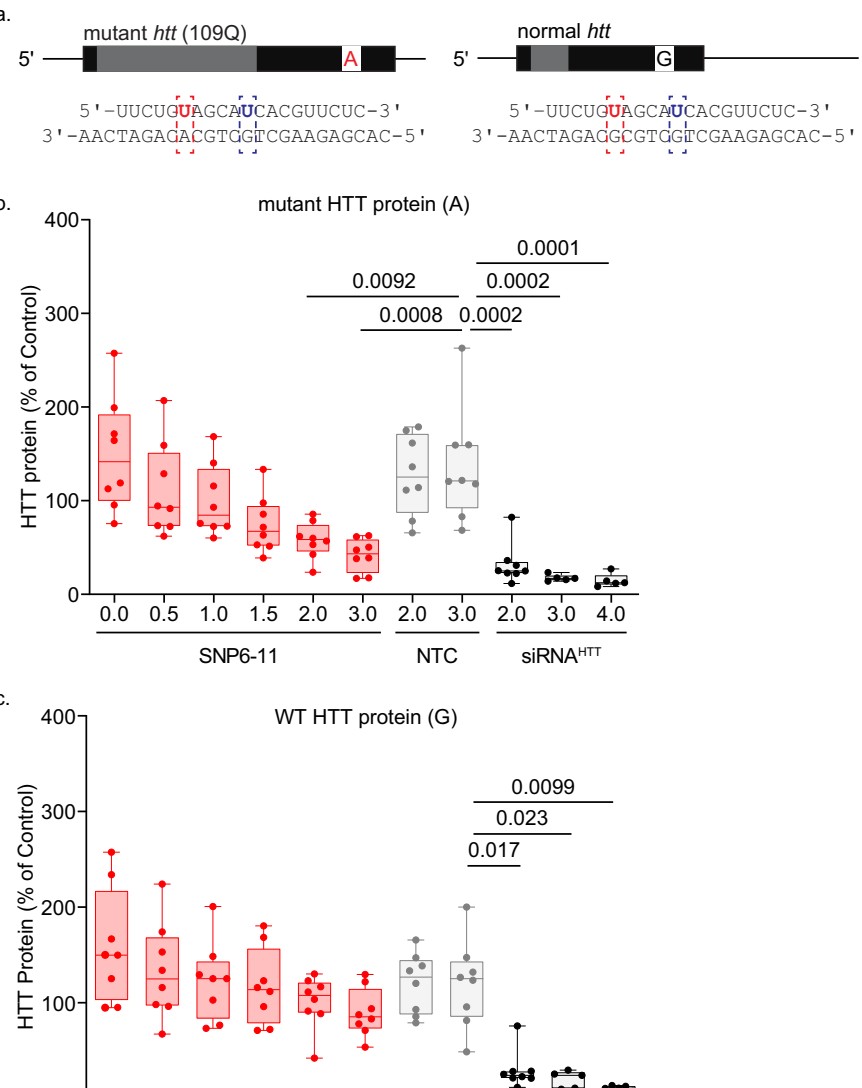

**Fig. 4 | SNP6-11 siRNA maintains selective-silencing of mutant HTT in patient-derived NSCs.** HD patient derived neural stem cells carrying an A or a G at SNP site rs362273 (**a**) were treated for 5 days with SNP6-11 or non-selective siRNA[HTT] at the concentrations listed. Graphs show pixel intensity quantification of mutant (**b**), or wild-type (**c**) HTT signal intensity standardized to that of vinculin reported as a percentage of untreated NSCs. A non-targeting control (NTC) siRNA was used as a negative control. Non-selective targeting with siRNA[HTT] was used as a positive control. One-way ANOVA and posthoc Tukey test; $n = 5$ wells for siRNA[HTT] at 3.0 and 4.0 μM, $n = 8$ wells per group for the remaining treatments. Error bars extend to minimum and maximum value. The lower bound of the box is the 25th percentile and the upper is the 75th, with a line at the median. Source data are provided as a source data file.

divalent siRNA chemical scaffold alone does not change the expression of either the wild type or the mutant huntingtin alleles (Fig. 6c, d). The animals treated with SNP6-11 showed >85% silencing ($P < 0.0001$) of the mutant protein with no change in levels of the wild-type HTT (Fig. 6c, d). Divalent siRNA accumulation in different brain regions varies[19], but selective silencing of mutant huntingtin protein is sustained throughout the brain, indicating that more than 50x discrimination in vitro is sufficient to support selective silencing in vivo. Selective mutant *htt* silencing did not change the levels of the neuron-specific protein Dopamine- and cAMP-regulated phosphoprotein (DARPP32; Supplementary Fig. 8a). Glial fibrillary acid protein (GFAP), a marker of astrocytes, was also unchanged, except in the striatum (Supplementary Fig. 8b). Levels of ionized calcium-binding adaptor molecule (Iba1), which localizes to microglia, were not different between treatment groups (Supplementary Fig. 9), indicating that exposure

to the divalent siRNA did not elicit significant microglial activation at the times tested.

To ensure that selective allele-specific silencing in vivo also occurs at a high siRNA dose, we injected the second cohort of animals with 450 μg (20nmol, 10 nmol/ventricle) of divalent siRNA, the maximum deliverable dose for this chemical class. Like the 225 μg injection, selective silencing of mutant *HTT* occurred throughout the brain (Supplementary Fig. 10). At one month, the time of greatest oligonucleotide accumulation, mutant HTT approached the lower detection limit (>99% silencing, $P < 0.0001$) with no decrease in wild-type huntingtin, indicating that SNP6-11 can support selective silencing at a wide range of doses throughout the brain over time. We expect this higher dose to maintain silencing for longer than 6 months[19].

Here we demonstrate that using a fully chemically modified siRNA, we can achieve selective silencing of mutant *HTT* in vivo, based

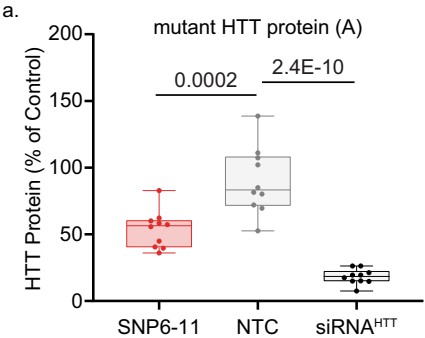

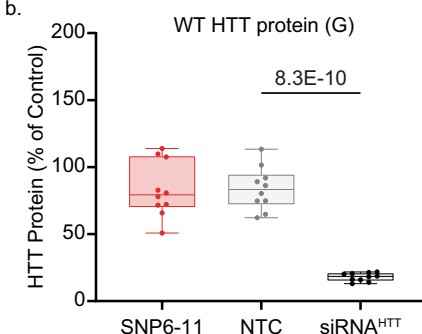

**Fig. 5 | SNP6-11A siRNA maintains selective-silencing of mutant HTT in neurons differentiated from human iPSCs.** Western blot analysis of HD109 neurons, which are heterozygous at SNP2273, showed lowering of mutant HTT **a** compared to wild-type HTT **b** with SNP6-11. Cells were treated for 5 days with 3.0 μM siRNA. Pixel intensity of mutant or wild-type HTT was standardized to that of vinculin and reported as a percentage of untreated. Non-selective targeting with siRNA[HTT] was used as a positive control. A non-targeting control (NTC) siRNA was used as a negative control. One-way ANOVA and posthoc Tukey test; $n = 10$ biological replicates (wells) per group. Error bars extend to minimum and maximum value. The lower bound of the box is the 25th percentile and the upper is the 75th, with a line at the median. Source data are provided as a source data file.

on a SNP heterozygosity. We expect this approach to be generalizable to other autosomal dominant disorders where expression of the wild-type allele is essential.

## Discussion

Allele-specific modulation of gene expression in the CNS promises to preserve normal function and enable disease-modifying treatment for many neurodegenerative disorders. Here, we demonstrate that fully chemically modified, therapeutically translatable siRNAs targeting SNP heterozygosities support allele-specific discrimination in human neurons derived from human patient iPSCs and in vivo in mice. Through repeated targeted screens and chemical optimization, we identified SNP-selective siRNAs that achieved more than 50× discriminative power in a cell-based assay and confirmed selective silencing of the mutant *HTT* allele (>85%) throughout the brain in a mouse model of HD. Sequence context and chemical composition affected selectivity; optimization for each SNP heterozygosity should be performed in the chemical configuration intended for in vivo use. Our findings provide a roadmap for identifying allele-specific therapeutic siRNAs for other disorders in which expression of the normal allele is essential and indicate that compounds with 50× discrimination in cell culture support robust allele-selective silencing in vivo.

For SNP-discriminating siRNAs, the sequence space available for screening is limited to approximately 16 bases immediately surrounding the SNP position. The placement of mismatches proved crucial for siRNA selectivity. For the rs362273 A isoform, mismatches in positions 6 and 11 showed the best potency to discrimination ratio. Whereas for rs362307 U isoform, positions 3 and 7 were optimal. These findings suggest that for maximum discrimination, the positioning of the SNP heterozygosity and of the second mismatch is dependent on the target sequence. Thus, the position, identity, and modification pattern of both primary and secondary mismatches need to be optimized individually for different SNP sites.

Although we identified SNP-selective siRNAs against both SNP sites evaluated, similar success might prove challenging for some SNP heterozygosities, particularly if the sequence around the SNP site is GC rich[62]. siRNAs vary vastly in their ability to silence their targeted mRNAs, with EC50 values ranging from the low pM to the low nM range. The efficacy of an siRNA is affected by the ability to enter RISC and the efficiency with which the loaded RISC complex can find the target, adopt the right conformation for cleavage, and release the resulting product[50,63]. As the sequence space available for screening of SNP-selective compounds is limited, their potency could be lower than the best non-selective compounds for the same target. For example,

the potency of both SNP-targeting compounds was less than siRNA[HTT], the previously identified non-selective HTT-targeting siRNA[13,19]. Consequently, therapeutic translation of SNP-selective compounds may require higher doses or more frequent dosing to achieve silencing comparable to their non-selective counterparts.

Chemical context near the SNP site affects selectivity. The introduction of 2′-F modifications increases the local oligonucleotide affinity[54]. Placing this modification near the SNP position increases the silencing of both alleles but reduces discrimination (Fig. 4). In contrast, 2′-OMe modifications reduce siRNA affinity to the target and, in many positions, are not sterically well tolerated. Manipulating affinity throughout the siRNA affected target discrimination. Introduction of bulky 2′-OMe modifications at mismatch sites increased discrimination while higher affinity 2′-F modifications decreased it. By optimizing sequence, structure, and chemical modification pattern, we identified multiple functional compounds with SNP-discriminating properties; however minor changes in the chemical modifications (e.g., a change from 2′-OMe to 2′-F in a single position) impacted both on-target activity and discrimination (Fig. 3). Therefore, optimization for efficacy and selectivity must be performed in the context of the modification pattern intended for in vivo use. Further exploration of an expanded range of chemical modifications by including, for example by bulkier entities like MOE (2′-methoxyethyl), or highly flexible UNAs (unlocked nucleic acid) might eliminate the need for structural mismatches.

The level of discrimination achieved in the cell-based assay with the rs362273 targeting SNP6-11A siRNA, (-58× difference in IC50 values) was sufficient for allele selective silencing in vivo in CNS. Treatment in mice achieved efficient silencing of mutant HTT protein in all brain regions without impact on wild-type HTT protein, despite varied siRNA accumulation throughout the brain. The thalamus accumulated -10–15× more siRNA than did the cortex (4,24). All brain regions demonstrated effective allelic huntingtin lowering. Nonetheless, the pharmacokinetic dose range of selective silencing in brain did not apply to non-brain tissues. The systemic administration of GalNac modified siRNA supports about a ten-fold higher accumulation in the liver compared to brain. The increased accumulation in liver dampened the selectivity of siRNA6-11. To recover oligonucleotide selectivity, introduction of an additional stereo-constrained vinyl phosphonate backbone next to the mismatch site was necessary to further enhance discrimination and maintain selectivity in vivo in the liver[64]. If the same accumulation in the brain as in the liver can be achieved, additional chemical enhancement of the siRNA might be necessary.

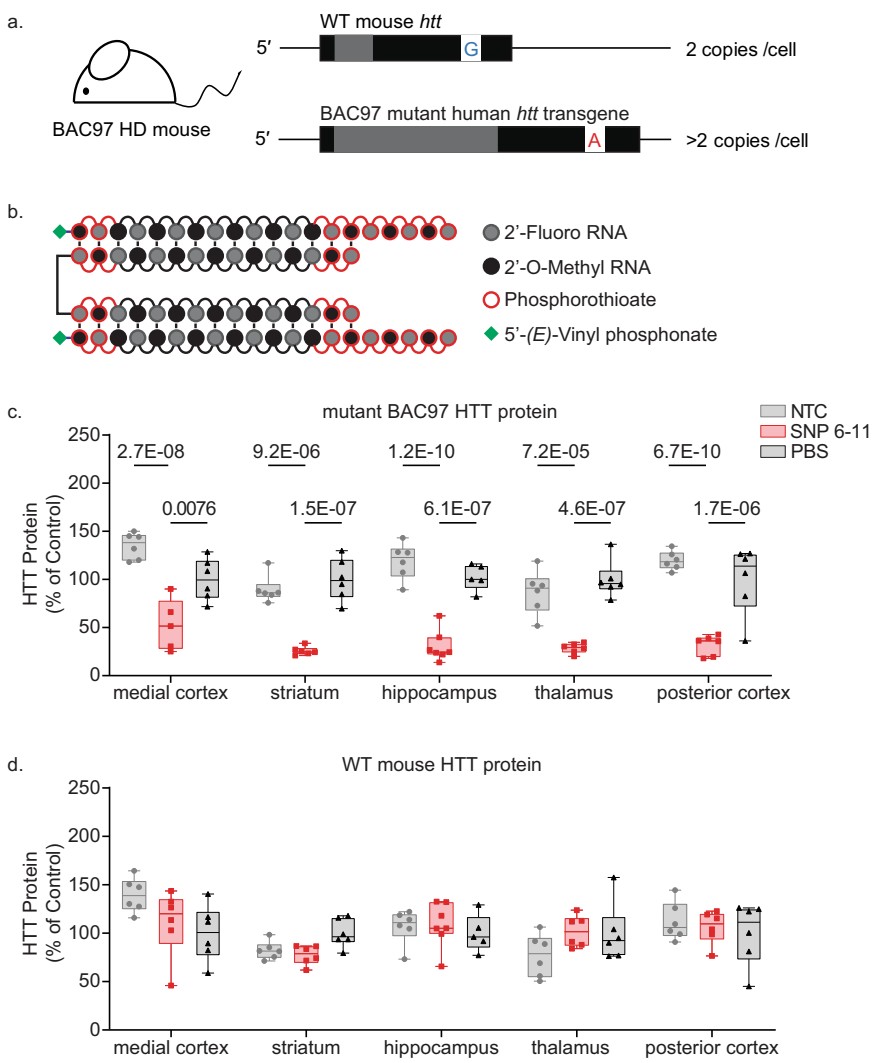

**Fig. 6 | SNP6-11 selectively silences mutant HTT in an HD mouse model. a** BAC97 HD mice have a high copy number of transgenic mutant human huntingtin (*Htt*) with a polyQ expansion of 97 repeats, and two normal copies of wild-type (WT) mouse *Htt*. Both the human BAC97 huntingtin and the WT mouse *Htt* have homology at the SNP6-11 target site, except for a heterozygosity at SNP site rs362273, mimicking the WT/mutant *Htt* SNP heterozygosity in patients. **b** for in vivo delivery, the SNP 6–11 compound was synthesized in a di-valent config-uration as depicted. **c** When treated with 225 μg (10nmols) of siRNA (SNP6-11) in a divalent scaffold, allele-specific silencing of mutant HTT protein in mouse brain is achieved one month after bilateral injection into the lateral ventricles (ICV injection). **d** Across brain regions, WT mouse HTT is preserved with no significant silencing detected (except in striatum), while transgenic mutant HTT is lowered significantly, averaging 70% silencing across brain regions (*p*-value <0.00001). Protein levels were measured via WES Protein Simple. A two-way ANOVA with multiple comparisons was performed for all protein analysis, comparing treatment groups to the PBS control for each brain region; *n* = 5–6 animals per group. Error bars extend to minimum and maximum value. The lower bound of the box is the 25th percentile and the upper is the 75th, with a line at the median. Source data are provided as a source data file.

In HD and other expanded repeat-associated diseases, the causative mutation is not amenable to direct targeting. Therefore, a panel of SNP-targeting compounds must be developed. Studies of the HD patient population and targetable SNPs[35,45,60,65] suggest that 80–85% of HD patients can be treated with panels of 2–4 SNP heterozygosities[34,35,60,65]. Up to 40% of patients of European descent are heterozygous at rs362273[35] and the A isoform is more common on HD chromosomes[61], we have successfully targeted the A isoform, but to treat 40% of patients both isoforms would need to be targeted. If the allele selective silencing of HTT becomes essential for clinical disease modification then the linkage between the CAG repeats and the SNP site heterozygosity would need to be established individually for eligible patients using long read sequencing[66].

## Methods

All experimental studies involving animals were approved by the University of Massachusetts Chan Medical School Institutional

Biosafety Committee (IBC) and Institutional Animal Care and Use Committees (IACUC; Protocol numbers A-2411, A-978). Stem cell experiments were performed with oversight of Human Embryonic Stem Cell Research Oversight Committee (ESCRO Committee) through the Partners (now MGB) Institutional Biosafety Committee (PIBC) at Partners Healthcare (ESCRO#: 2015-01-02 and PIBC Reg# 2017B000023).

### Cell treatment: reporter assay

HeLa cells (ATCC CCL-2) were grown and maintained in DMEM (Gibco ref.# 11965-092) with 1% penicillin/streptomycin and 10% heat inactivated fetal bovine serum (FBS). Three days prior to treatment, two 10 cm² dishes are plated with 2 × 10⁶ HeLa cells and grown overnight. The following day, DMEM is replaced with OptiMEM (Gibco ref.# 31985-070) and 6ug of reporter plasmid[35] is added to cells using lipofectamine 3000 (Invitrogen ref.# L3000-015), following the manufacturer's protocol. Cells are left in the OptiMEM/lipofectamine

overnight to allow for maximum reporter plasmid transfection. The following day, cells are ready for treatment.

For screening purposes, siRNAs are prepared in OptiMEM at 3 μM concentration. 50 μL of 3 μM siRNA is then added to 96-well white wall clear bottom tissue culture plate, in triplicate, for each reporter plasmid. For a dose response curve, 800 μL of 3 μM siRNA is prepared in OptiMEM, and a 1:1 dilution series is performed six times for seven concentrations of each siRNA: 3 μM, 1.5 μM, 0.75 μM, 0.375 μM, 0.1875 μM, 0.09375 μM, 0.046875 μM. 50 μL of each siRNA dilution are then added to each white wall clear bottom 96 well plate in triplicate. After the desired amounts of siRNA were added to the plate, 50 μL per well of HeLa cells transfected with reporter plasmids were added, resuspended in DMEM with 6% heat inactivated FBS at $0.15 \times 10^6$ cells/mL, bringing maximum final siRNA concentration to 1.5 μM.

After 72 h of treatment (100% confluency) cells were washed twice with PBS, then lysed with 20 μL 1× passive lysis buffer from Promega dual-luciferase assay system kit (Promega ref.# E1960) and placed on the shaker for 15–20 min. Luminescence was read after adding 50 μL Luciferase Assay Reagent II (Promega kit), then read a second time after the addition of 50 μL/well of stop and glow reagent. Luminescence values were normalized to untreated controls and graphed on a log scale.

## Stem cells

Experiments were performed with oversight of Human Embryonic Stem Cell Research Oversight Committee (ESCRO Committee) through the Partners (now MGB) Institutional Biosafety Committee (PIBC) at Partners Healthcare (ESCRO#: 2015-01-02 and PIBC Reg# 2017B000023). Human induced pluripotent stem cells (iPSCs) CS09iCTR-109n4 (CS vial ID: 1034860) and CS09iCTR-109n5 (CS vial ID: 1034589) hereafter referred as HD109 (clone n4 and n5) described by (Mattis VB et al., 2015)[67], were acquired from Cedars-Sinai Regenerative Medicine Institute Induced Pluripotent Stem Cell Core. Cells were grown in mTeSR™ Plus kit (StemCell Technologies, 100-0276) on plastic plates coated with Matrigel (Corning, 354277) Cell lines were karyotyped after in-house expansion by Cell Line Genetics using array comparative genomic hybridization (aCGH, Agilent 60 K Standard). Limited chromosomal imbalances in non-critical genes were reported: a single deletion of 460 kb in chromosome 2, cytoband q22.1 (*LRP1B*) in HD109 n4, while HD109 n5 displayed an additional 90 kb deletion in chromosome 14, cytoband 24.3 (*SPTLC2*). The DNA sequence of HTT at SNP RS362273 (hereafter named SNP2273) was determined via Sanger sequencing by Genewiz, Inc. using primers previously published[35] and shown to be heterozygous (G/A) for HD109.

## Neural stem cells (NSCs)

NSCs were differentiated as described[68]. NSCs were maintained in NSC Medium without 1% KnockOut serum replacement and plated on 48-well (Corning, 3548) plastic plates coated with 0.1% Gelatin for siRNA experiments.

## siRNA treatment of NSCs and Neurons

For NSCs, normal growth medium was replaced with NSC medium without antibiotics and containing cholesterol-tagged siRNAs targeting total HTT (HTT10150), allele specific HTT siRNAs, or a non-targeting control (NTC) diluted to indicated 1× in 48 well plates. NSCs were treated for 5 days with a complete medium change at day 3 with medium containing fresh siRNAs at 1× concentration. For human neuron cultures in 48 well plates, on differentiation day 42 NMM was switched 250 μl/well to NMM without antibiotics/mitotic inhibitors 24 h prior to treatment. On Day 0 of treatment (differentiation day 43), 200 μL of NMM was removed from each well leaving 50 μl remaining, and 250 μL/well of complete medium containing siRNAs at 2× concentrations were introduced bringing the volume to 300 μl (0.83 dilution factor;

final concentration of siRNAs, 1.66×). After 24 h of treatment, additional 200 μL of NMM without antibiotics/mitotic inhibitors was added to each well bringing the volume to 500 μl (0.6× dilution factor; final concentration of siRNAs, 1×) and incubated for 4 more days (total 5 day treatment). Concentrations indicated in Fig. 4 those of 4 day incubation (1×). 30 ul of lysis buffer (50 mM Tris, pH 7.4, 250 mM NaCl, 1% Nonidet P-40, 5 mM EDTA) containing protease inhibitors per well was used to lyse cells for western blot analysis.

## Animal compliance and models

All experimental studies involving animals were approved by the University of Massachusetts Medical School IACUC Protocols (#A-2411, A-978) and performed according to the guidelines and regulations therein described. Briefly, mice were housed with a maximum of 5 per cage in a pathogen-free facility under standard conditions with access to food, water, and enrichment *ad libitum*.

BAC97 HD[44] male mice were bred with FVB/NJ (The Jackson Laboratory Strain #001800) female mice to produce mixed litters with -50% heterozygous mutant offspring. The litters were weaned, and ear punches were genotyped via PCR to validate the presence of the BAC97 transgene.

## Stereotactic brain injections into the lateral ventricle

At approximately 8 weeks of age, BAC97 females were anesthetized with tribromoethanol at 284 mg/kg weight. Mice were subsequently placed in a stereotaxis and hair was removed from scalp using glue. The injection site was cleaned and prepared for surgery with three applications of betadine and ethanol swabs. An incision was made at the top of the scalp and the bregma was located as a reference point for injection sites. The needle was placed +/−0.8 mm mediolateral and 0.2 mm posterior to the bregma on both sides of skull for bilateral injection. Holes were drilled and the needle was lowered 2.5 mm for placement into the lateral ventricle. After one minute, 5 μL of 2000 μM (*n* = 12) or 4000 μM (*n* = 12) siRNA and NTC siRNA, suspended in PBS was injected into the right lateral ventricle at a rate of infusion of 500 nL/min, followed by an identical injection into the left ventricle, for a total injection of 10 μL/40nmols siRNA. The procedure was followed for PBS vehicle control injections (*n* = 12). Needles were left in place for one minute following the end of infusion to avoid backflow. Upon removal from stereotaxis, the incision was closed with staples and Meloxicam SR was administered for analgesia. Mice recovered on a heating pad until awake and sternal.

Tissues were harvested one-month post-injection. Mice were euthanized with isofluorane and brains were removed and cut into 300 micron sections using a vibratome submerged in PBS at -0 °C. Sections were suspended in PBS on ice, where 2 mm punches were taken using a sterile disposable biopsy punch (Integra Militex REF#33-31-P/25) from 3 sections bilaterally for each brain region: striatum, medial cortex, posterior cortex, hippocampus, and thalamus. Punches from one side of the brain were flash frozen in liquid nitrogen for subsequent protein analysis. Punches from the opposite side of the brain were saved in RNA later for mRNA analysis via branched DNA assay.

## Quantigene 2.0 branched DNA assay

Tissue punches were removed from RNA later and lysed with sterile beads in 300 μL/sample of homogenizing solution (Quantigene sample processing kit, Invitrogen CAT#QG0517) and 6 μL/sample of 20 mg/mL proteinase K (Qiagen, mat #1114886). Lysis completed with a 30 min incubation at 55 °C. Lysates were removed from beads and stored at −80 °C. On day of analysis, lysates were thawed and again incubated at 55 °C for 30 min. Quantigene 2.0 probe sets prepared for human *HTT* (SA-50339) mRNA, mouse *Htt* (SB-14150) mRNA, and for mouse *HPRT* (SB-15463) mRNA as a housekeeping gene. 60 μL of each probeset (mixture of mRNA specific probe, blocking reagent, Quantigene 2.0

lysis mixture, and water) was added to Quantigene plates (PSCP/HV DNA coated wells REF# 15553) in addition to 40 μL of lysate. Quantigene plates were subsequently incubated overnight at 55 °C.

On the following day, the plates were washed three times with wash buffer (Quantigene REF# 10843, and REF# 10846) and 100 μL/well of pre-amplifier (Quantigene REF# 15095) was added to the plate, which was then incubated for one hour at 55 °C. The wash step was repeated and 100 μL/well of amplifier (Quantigene REF# 15098) was added, and the 1 h incubation was repeated. Following incubation, the plates were washed again, and 100 μL/well of label probe (Quantigene REF# 13241) was added. Plates were then incubated for one hour at 50 °C. The wash step was repeated one last time, and 100 μL/well of substrate (Invitrogen REF# 144558) was added. Luminescence was measured within 15 min.

Background signal was subtracted, and all reads were normalized to PBS-injected control. Two-way ANOVA with Tukey's multiple comparisons test was performed using GraphPad Prism software. The significance is relative to the PBS injected control.

## Western blot and WES
Frozen tissue punches were homogenized on ice in 75 μl 10 mM HEPES pH 7.2, 250 mM sucrose, 1 mM EDTA + protease inhibitor tablet (Roche) + 1 mM NaF + 1 mM Na3VO4, sonicated for 10 s and protein concentration determined by Bradford assay (BioRad). For western blot, equal amounts of protein (2.5 or 10 μg) were separated by SDS-PAGE on 4–12% Bis-Tris or 3–8% Tris acetate gels (Criterion XT, BioRad) and transferred to nitrocellulose using TransBlot Turbo (BioRad). Blots were probed with antibodies to HTT (aa1-17, Ab1[69]), GFAP (Millipore; Cat# AB5804, Lot# 3538088; ProteinSimple, 1:3000), DARPP32 (abcam; Cat# ab40801, Clone EP720Y, Lot# GR3213231-12; ProteinSimple, 1:2,000), Iba1 (Fujifilm; Cat# 019-19741, Lot# CAG5175; WB, 1:500) and with antibodies to loading controls Vinculin (Sigma; Cat# V9131, Clone hVIN-1, Lot# 036M4797V; WB, 1:2,000; ProteinSimple, 1:5000) and GAPDH (Millipore; Cat# MAB374, Clone 6C5, Lot# 3527693; WB, 1:10,000). Bands were visualized with SuperSignal West Pico PLUS Chemiluminescent Substrate (Thermo) and Hyperfilm ECL (GE Healthcare). Total signal intensity was determined using the scanned films by manually tracing the bands in ImageJ (v. 1.53 s) software (NIH) and multiplying the area by the average signal intensity.

We also analyzed the same protein samples using the Wes system (ProteinSimple). The standard settings for the 66–440 kDa separation modules were used with 0.2 mg/ml of each lysate and anti-HTT (Ab1, 1:50) plus anti-Vinculin (Sigma, 1:2000) antibodies. Peak area was determined using Compass for SW software (ProteinSimple) and dropped line fitted peaks.

Two-way ANOVA with Tukey's multiple comparisons test was performed on the normalized signal intensity and normalized peak area using GraphPad Prism software. Normalization and significance were calculated relative to the PBS injected control.

## Thermo stability assay
Briefly, 1 μM guide strand and 1 μM complementary sense strand were annealed in a 10 mM sodium phosphate buffer (pH 7.2) containing 100 mM NaCl and 0.1 mM EDTA by heating at 95 °C for 1 min and cooled down gradually to room temperature. Tm measurement was performed with temperature controller. Both the heating and cooling curves were measured over a temperature range from 20 to 95 °C at 1.0 °C/min three times[64].

## Oligonucleotide synthesis
Sequences and chemical modifications of oligonucleotides are provided in Supplementary Tables 1 and 2. Oligonucleotides were synthesized by phosphoramidite solid-phase synthesis on a Dr Oligo 48 (Biolytic, Fremont, CA), or MerMade12 (Biosearch Technologies, Novato, CA), using 2′-F or 2′-OMe modified phosphoramidites with standard protecting groups. 5′-(E)-Vinyl tetra phosphonate (pivaloyloxymethyl) 2′-O-methyl-uridine 3′-CE phosphoramidite (VP) was purchased from Hongene Biotech, USA. All other phosphoramidites used were purchased from ChemGenes, Wilmington, MA. Phosphoramidites were prepared at 0.1 M in anhydrous acetonitrile (ACN), except for 2′-O-methyl-uridine phosphoramidite dissolved in anhydrous ACN containing 15% dimethylformamide. 5-(Benzylthio)−1H-tetrazole (BTT) was used as the activator at 0.25 M and the coupling time for all phosphoramidites was 4 min. Detritylations were performed using 3% trichloroacetic acid in dichloromethane. Capping reagents used were CAP A (20% n-methylimidazole in ACN) and CAP B (20% acetic anhydride and 30% 2,6-lutidine in ACN). Reagents for capping and detritylation were purchased from American International Chemical LLC (AIC), Westborough, MA. Phosphite oxidation to convert to phosphate or phosphorothioate was performed with 0.05 M iodine in pyridine-H2O (9:1, v/v) (AIC) or 0.1 M solution of 3-[(dimethylaminomethylene)amino]−3H-1,2,4-dithiazole-5-thione (DDTT) in pyridine (ChemGenes) for 4 min. Unconjugated oligonucleotides were synthesized on 500 Å long-chain alkyl amine (LCAA) controlled pore glass (CPG) functionalized with Unylinker terminus (ChemGenes). Cholesterol conjugated oligonucleotides were synthesized on a 500 Å LCAA-CPG support, where the cholesterol moiety is bound to tetra-ethylenglycol through a succinate linker (Chemgenes, Wilmington, MA). Divalent oligonucleotides (DIO) were synthesized on modified solid support[19].

## Deprotection and purification of oligonucleotides for screening of sequences
Prior to the deprotection, synthesis columns containing oligonucleotides were treated with 10% diethylamine (DEA) in ACN to deprotect cyanoethyl groups. In synthesis columns, both unconjugated and cholesterol conjugated oligonucleotides on solid support were then deprotected with monomethylamine gas (Specialty Gases Airgas) for an hour at room temperature. Deprotected oligonucleotides released from the solid support were precipitated on the support by passing solution of (i) a mixture of 0.1 M sodium acetate in 85% ethanol and then (ii) 85% ethanol to the synthesis column. The excess ethanol on solid support was dried by air flow and the oligonucleotides were flushed out by passing water through the column. This procedure renders pure oligonucleotides used for in vitro experiments.

## Deprotection and purification of oligonucleotides for in vivo experiments
Prior to the deprotection, oligonucleotides on solid support were treated with 10% DEA in ACN in synthesis columns to deprotect cyanoethyl groups. Divalent oligonucleotides (DIO) were cleaved and deprotected by AMA treatment for 2 h at 45 °C. The VP containing oligonucleotides were not treated with DEA post-synthesis and were cleaved and deprotected as described previously[70]. Briefly, CPG with VP-oligonucleotides was treated with a solution of 3% DEA in 28–30% ammonium hydroxide at 35 °C for 20 h.

All solutions containing cleaved oligonucleotides were filtered to remove the CPG and dried under vacuum. The resulting pellets were re-suspended in 5% ACN in water. Purifications were performed on an Agilent 1290 Infinity II HPLC system. Crude oligonucleotides were purified using a custom 30 × 150 mm column packed with Source 15Q anion exchange media (Cytiva, Marlborough, MA); running conditions: eluent A, 10 mM sodium acetate in 20% ACN in water; eluent B, 1 M sodium perchlorate in 20% ACN in water; linear gradient, 10–35% B in 40 min at 50 °C. Flow was 30 mL/min, peaks were monitored at 260 nm. Fractions were analyzed by liquid chromatography mass spectrometry (LC–MS). Pure fractions were combined and dried under vacuum. Oligonucleotides were re-suspended in 5% ACN and desalted by size exclusion on a 50 × 250 mm custom column, packed with Sephadex G-25 media (Cytiva, Marlborough, MA), and lyophilized.

## LC–MS analysis of oligonucleotides

The identity of oligonucleotides was verified by LC–MS analysis on an Agilent 6530 accurate mass Q-TOF using the following conditions: buffer A: 100 mM 1,1,1,3,3,3-hexafluoroisopropanol (HFIP) and 9 mM triethylamine (TEA) in LC–MS grade water; buffer B:100 mM HFIP and 9 mM TEA in LC–MS grade methanol; column, Agilent AdvanceBio oligonucleotides C18; linear gradient 0–35% B 5 min was used for VP and DIO oligonucleotides); linear gradient 50–100% B 5 min was used for TegChol conjugated oligonucleotides; temperature, 60 °C; flow rate, 0.85 ml/min. LC peaks were monitored at 260 nm. MS parameters: Source, electrospray ionization; ion polarity, negative mode; range, 100–3200 m/z; scan rate, 2 spectra/s; capillary voltage, 4000; fragmentor, 200 V; gas temp, 325 °C. Deprotection, purification, and LC-MS reagents were purchased from Fisher Scientific, Sigma Aldrich, and Oakwood Chemicals.

## Statistics and reproducibility

We have previously established and published that $n = 6$–10 mice per group produces robust data. Mice were assigned randomly to treatment groups. In vivo (mouse) studies and studies in human NSCs were repeated twice. In vivo studies have subsequently been replicated in alternative mouse models. For protein analysis, data were excluded if signal for the housekeeping gene was absent. Statistical tests were as described in the figure legends and text. Experimenters were not blinded to the treatment groups when performing the analyses.

## Reporting summary

Further information on research design is available in the Nature Research Reporting Summary linked to this article.

## Data availability

Source data are provided with this paper. Raw data is stored on internal dropbox folders and will be provided upon request. Any LC-MS data not included in source data is also available upon request. Source data are provided with this paper.

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

## Acknowledgements
Support for this study was provided by NS104022, CHDI Research Agreements #A-5038 and #A-6119, and the Berman-Topper Fund. Oligonucleotide synthesis was supported by S10 OD020012. Protein analysis was supported by NIH U01 NS114098, the CHDI Foundation, Award #A-6367 and the Dake Family Fund.

## Author contributions

E.L.P., N.A., and A.K. conceived the study. F.C., E.L.P., J.F.A., A.K., K.K.G., and M.D. designed the experiments. M.R.H., D.E., J.S., and K.Y. designed and synthesized oligonucleotides. R.M., B.M.D.C.G., E.G.K., E.S., F.M., and A.B. performed the experiments. F.C., R.M., E.L.P., E.S., and K.K.G. analyzed experiments and generated the figures. The manuscript was written by F.C., E.L.P., N.A., and A.K. with help from K.K.G. and M.D. and reviewed and edited by all the authors.

## Competing interests

The authors declare the following competing interests: F.C., E.L.P., J.F.A., K.Y., N.A., and A.K. are inventors on patent application #16/988,391 based on the results reported in this work; the applicant is the University of Massachusetts, the status of which is currently non-final rejection. The basis of this application is contained in Figs. 1–3. N.A. and E.P. are inventors on patent #20120136039 assigned to the University of Massachusetts upon which Fig. 2 is based and A.K. and J.A. are inventors on patent #20200087663, assigned to the University of Massachusetts, upon which the chemical modifications patterns used in this study are based. A.K. and N.A. are co-founders and own stock in Atalanta Therapeutics. M.R.H. and B.M.D.C.G. are now employees of Atalanta Therapeutics. All other authors declare no competing interests.

## Additional information

**Supplementary information** The online version contains

supplementary material available at

