## [Peer Review File · Nature Communications]

Chemical engineering of siRNAs for allele-specific gene silencing in vivo in CNSREVIEWER COMMENTS

Reviewer #1 (Remarks to the Author):

The manuscript by Conroy and colleagues describes the optimization of an siRNA that selectively inhibits the mutant huntingtin (HTT) mRNA and preserves the expression of unaffected allele. Allele selective targeting of the mutant HTT RNA by various RNA targeting strategies has been well described in the literature, including by the authors of this manuscript. The authors described further optimization of two different siRNA agents for greater discrimination of mutant versus "wild-type) HTT mRNA and use of newer siRNA designs that enhance in vivo activity. Because this is largely an optimization of previous concepts, it is better suited for a journal specializing in oligonucleotide therapeutics where these improvements will be better appreciated by the target audience.

Some additional points for the authors to consider:

- 1. All the optimization and selectivity data appear to be done using a reporter assay. As there are large numbers of patient derived fibroblast cells available that have been well-characterized for the haplotypes of both HTT alleles, it would add to the work to show that there is similar selectivity for the HTT RNA species in native context. A dose response study in the fibroblasts using the optimized siRNA RNA designs would add to the manuscript.**
- 2. It would be useful to compare the potency of the optimized allele selective siRNA to the non-allele selective siRNA previously published by this group in fibroblast cells. Does limiting the siRNA to the SNP sites target result in equi-potent siRNAs?**
- 3. Do one of the fully humanized HTT mouse models contain the appropriate SNP pairs to evaluate selectivity in vivo? Although the siRNA does recognize the mouse htt transcript, there are significant sequence differences between human and mouse HTT genes that could have non-obvious effects on potency of the ASOs.**
- 4. Suggest also comparing against the non-allele selective siRNA published in the Alterman et al. manuscript in the mouse model.**
- 5. It is not possible to calculate fold selectivity based on a single dose. Suggest doing a dose response in the mouse model.**
- 6. The upregulation of the normal mouse htt transcript warrants further investigation/discussion. This does not appear to occur when the mutant htt RNA is reduced with RNase H oligonucleotides. Does this occur at the RNA level as well? Does it occur in human cells? Cortical neurons from the transgenic mice? This effect seems to be most prevalent in striatum where the authors also reported evidence of siRNA-induced astrogliosis. Is this related to a toxicity? Does it occur with a non-allele selective siRNA that only targets the human transgene?**
- 7. For the mouse studies- is the listed dose the total dose administered or dose per ventricle?**
- 8. Did the authors perform a histological analysis of brain tissue to see if there are any notable histological changes in striatum that correlate with increased GFAP? Was AIF1 measured?**
- 9. Should reference the original BAC97 paper in methods.**
- 10. Reference 18 and 45 appear to be duplicates.**

Reviewer #2 (Remarks to the Author):

The manuscript "Chemical engineering of siRNAs for allele-specific gene silencing in vivo in CNS" by Conroy et al. makes use of mismatch mediated selectivity of siRNAs for Huntington disease model. The concepts and rationale have been well-thought out and has been done in a logical manner.

Before this article is accepted for publication, the following questions need to be addressed:

- 1. The authors state "chemical and thermodynamic mediated optimization" but there is no experiment or data to explore chemical modification and mismatch related**

thermodynamic binding affinity. The authors need to measure the thermodynamic melting temperatures for the chemically modified RNA strands against their RNA target sequences. Such a method has been shown in literature for this gene before and it is a doable experiment. This will explain

- a. why the selectivity changes occur in going from one sequence to the other,
- b. why double mismatch works better than a single mismatch and
- c. why certain mismatch base pairs (e.g., C:U vs G:U) work better than others
- d. why 6-11 double mismatch works better than other mismatches

2. The oligonucleotide synthesis details should include whether the duplexes have regular phosphates or E-vinyl phosphonates at the 5'-end, where the PS modifications are and if they have a lipophile. A simple Table showing duplexes with chemical modifications will address this problem.

Such confusion arises because in p-4 it is stated

"All terminal backbones were modified with phosphorothioates (PS) (Figure 1a). - When cholesterol-conjugated, this type of compound is readily internalized by many cell types in vitro. The same chemical scaffold can later be re-synthesized in a divalent configuration for delivery to the CNS, allowing seamless transition between in vitro and in vivo experimental systems.

" whereas Figure 1 does not show full P=S and does not show cholesterol conjugate.

Also, discuss the biodistribution differences between PS backbone RNA vs cholesterol conjugates.

3. The in vitro experiments have been done with monovalent siRNAs whereas the in vivo testing is done with divalent siRNAs: is there a correlation of monovalent to divalent siRNAs in vitro? Why the divalent molecules were not tried for in vitro experiments?

4. Again, clearly show the structure of the divalent siRNAs used for the in vivo experiment in Figure 5.

a. Does it have a phosphate or vinylphosphonate?

b. Does it have cholesterol molecules or just P=S linkages?

c. Also explain, Why a control divalent siRNA is not used in this in vivo experiment

5. Explain the 2'-OMe vs 2'-F based activity differences based on Ago2 loading, thermodynamics and relative metabolic stability of the siRNAs used. Metabolic stability of OMe vs F construct should be evaluated in mouse brain tissues.

6. Explain why ICV is used for in vivo experiment? This is not a viable method for potential therapeutic applications. Has this experiment been tried by intrathecal (IT) dosing? Does the divalent molecule reach different parts of brain by IT administration? Does it show the same selectivity?

7. References 36 and 37 are the same.

8. References 18 and 45 are the same.

Reviewer #3 (Remarks to the Author):

This manuscript by Khvorova and colleagues explores possible variations of htt-targeting siRNA design that can render these SNP-based, allele-targeted siRNAs more allele-specific. This is not only an important elucidation of modifications allowing selective targeting mutant versus wild-type htt alleles, but also an excellent contribution to the design of SNP-targeted siRNAs in general. The former consideration is important and timely given the high interest in clinical application of htt-targeting gene therapies for Huntington's disease. In addition, the latter renders the article of high general interest with regard to the rationale design of SNP-targeted gene therapies in general.

The manuscript is concise and well-written. The conclusions are solid overall and these will have important gene therapy applications.

I was also pleased to read the authors' careful consideration of the upregulation of the endogenous, WT allele in parallel with the reduction in mutant HTT protein.

As a very minor revision, I would like to see the authors include further reflection on the possible significance of the non-sequence-specific effect on the upregulation of GFAP in an important Huntington's disease-affected brain region (striatum). At the authors' discretion, I would also welcome further experimental analyses to establish whether or not this might have been accompanied by other signs of gliosis or inflammation.

Overall, this is an excellent study that will contribute meaningfully to the previous literature in the fields of Huntington's disease therapies, and design of siRNA-based therapeutic agents in general.

The relevance of these findings to current gene therapy strategies for counteracting the neuropathogenesis of mutant HTT warrants particularly rapid publication.

REVIEWER COMMENTS

Please enclosed find the revised manuscript “**Chemical engineering of therapeutic siRNAs for allele-specific gene silencing *in vivo* in CNS.**” We apologize for the significant delay in the resubmission of the review, which was in part caused by COVID-related operational and staff disruptions.

At this point, this article still represents the first ever demonstration of siRNA-based allele selective silencing *in vivo*. This is potentially even more clinically significant with the recent clinical failure of pan-targeting huntingtin approaches.

We highly appreciate the productive feedback from the reviewers and have addressed their comments in full. The detailed response is provided below. The new data and figures are incorporated within this response for the reviewer’s convenience.

The major additional data comprises the demonstration of selective silencing in patient derived neural stem cells and differentiated neurons. We believe this represents a significant addition to the paper, demonstrating allele specificity in a fully human context and in a relevant cell type. We have also included a comparison of the efficacy of SNP targeting compounds to non-selective compounds, and the evaluation of the impact of a mismatch on oligonucleotide Tm.

Reviewer #1 (Remarks to the Author):

*The manuscript by Conroy and colleagues describes the optimization of an siRNA that selectively inhibits the mutant huntingtin (HTT) mRNA and preserves the expression of unaffected allele. Allele selective targeting of the mutant HTT RNA by various RNA targeting strategies has been well described in the literature, including by the authors of this manuscript. The authors described further optimization of two different siRNA agents for greater discrimination of mutant versus “wild-type) HTT mRNA and use of newer siRNA designs that enhance *in vivo* activity. Because this is largely an optimization of previous concepts, it is better suited for a journal specializing in oligonucleotide therapeutics where these improvements will be better appreciated by the target audience.*

While the concept of SNP-based discrimination has been around for quite some time, this article is the first-ever demonstration of productive RNAi-based allele-specific modulation of gene expression *in vivo* in the CNS. The article proves that this is feasible and defines the framework (with pros and cons) for others to follow. With the recent failure of clinical trials of ASOs for Huntington’s disease, we believe this also presents an alternative path forward for HD therapeutics.

Some additional points for the authors to consider:

1. *“All the optimization and selectivity data appear to be done using a reporter assay. As there are large numbers of patient derived fibroblast cells available that have been well-characterized for the haplotypes of both HTT alleles, it would add to the work to show that there is similar*

selectivity for the HTT RNA species in native context. A dose-response study in the fibroblasts using the optimized siRNA RNA designs would add to the manuscript.”

Thank you. As the patient-derived fibroblasts can have very low expression of huntingtin, we have added data demonstrating selective silencing of the mutant allele in patient derived neural stem cells and in neurons differentiated from patient-derived iPSCs. The figures and additional text are provided below.

Figure 4. SNP6-11A siRNA maintains selective-silencing of mutant HTT in patient-derived NSCs HD patient derived neural stem cells carrying an A or a G at SNP site rs362273 **(a)** were treated for 5 days with SNP6-11 or non-selective siRNA^{HTT} at the concentrations listed. **(b,c)** Graphs show pixel intensity quantification of mutant **(b)**, or wild-type **(c)** HTT signal intensity standardized to that of vinculin reported as a percentage of untreated NSCs. A non-targeting control (NTC) siRNA was used as a negative control. Non-selective targeting with siRNA^{HTT} was used as a positive control. ANOVA and posthoc Tukey test. Error bars, SD; N=5-8.

Figure 5. SNP6-11A siRNA maintains selective-silencing of mutant HTT in neurons differentiated from human iPSCs. Western blot analysis of HD109 neurons, which are heterozygous at SNP2273, showed lowering of mutant HTT **(a)** compared to wild-type HTT **(b)** with SNP6-11. Cells were treated for 5 days with 3.0 μ M siRNA. Pixel intensity of mutant or wild-type HTT was standardized to that of vinculin and reported as a percentage of untreated. Non-selective targeting with siRNA^{HTT} was used as a positive control. A non-targeting control (NTC) siRNA was used as a negative control. ANOVA and posthoc Tukey test, n=10.

“Allele-selective lowering of mutant HTT protein in human neural stem cells and in stem cells differentiated into neurons

To determine if SNP6-11 could discriminate between wild-type and mutant huntingtin in a native human context, we treated human neural stem cells (NSCs) with SNP6-11 siRNA. The NSCs were created from induced pluripotent stem cells (iPSCs) with 109 CAG repeats. We performed DNA sequencing and determined that these cells were heterozygous at rs362273. There are three major huntingtin haplogroups, A, B and C⁶². A majority of HD chromosomes belong to the A haplogroup, representing targetable SNPs associated with HD. In contrast, the C haplogroup is overrepresented on normal chromosomes⁶². Evidence from other SNP heterozygosities suggests that the parental CAG109 cell line belonged to the A or B haplogroups⁶³⁻⁶⁵. Both A and B haplogroups have an A on the mutant isoform at the rs362273

SNP site; since we had established heterozygosity, we reasoned that the non-HD chromosome would belong to haplotype C and have a G at the SNP site.

After treatment with siRNA, we measured the levels of wild-type and mutant HTT protein by SDS-PAGE and western blot. The non-selective siRNA^{HTT} lowered both wild-type and mutant HTT protein (Figure 4a). In contrast, when compared to a non-targeting control (NTC) at 3.0 μ M, SNP6-11 produced a dose-responsive lowering of mutant HTT at 1.5, 2.0, and 3.0 μ M (Figure 4a) with no corresponding reduction of wild-type HTT (Figure 4a). This data indicates that based on a single nucleotide difference and in the presence of two native human *htt* alleles, the SNP 6-11 siRNA selectively lowers mutant HTT (A) without affecting the wildtype protein (G).

Next, we asked if SNP6-11 could reduce mutant HTT protein selectively in a disease-relevant cell type. HD109 iPSCs were differentiated to cortical neurons. Western blot analysis showed that the neuronal marker β III-tubulin was present in cell lysates and immunofluorescent labeling of parallel wells revealed a high percentage of β III-tubulin positive cells, confirming neuronal enrichment (Supplementary Figure 6a,b). siRNAs were added for 5 days continuously, starting on differentiation day 43, as described in Methods. Compared to NTC at 3.0 μ M, SNP6-11 at 2.0 and 3.0 μ M significantly lowered mutant HTT protein by 35 and 38%, respectively (Figure 5a). In contrast, levels of wild-type HTT protein were unchanged (Figure 5b). At 3.0 μ M, non-selective siRNA^{HTT} lowered both mutant and wild-type HTT protein by 80% and 78% (Figure 5a,b). Like in HeLa cells (Supplementary Figure 2), the SNP 6-11 siRNA shows efficient discrimination between the two alleles but lower efficacy when compared to the non-selective siRNA^{HTT}.”

2. It would be useful to compare the potency of the optimized allele selective siRNA to the non-allele selective siRNA previously published by this group in fibroblast cells. Does limiting the siRNA to the SNP sites target result in equi-potent siRNAs?

New supplementary figure 6 compares the efficacy of potent pan-HTT silencing siRNA and SNP-selective siRNA in fibroblasts. We believe that the reviewer raised a very important point, as a reduction in the targetable sequence space does result in a reduction in potency. To accurately estimate the loss in potency, we also compared the dose-response of SNP-targeting and the best pan-HTT targeting clinical-stage siRNAs in HeLa cells. We observe 2-4 fold loss of efficacy. The new figure and the additional text from the results and discussion sections are below

Supplementary Figure 2. SNP-targeting siRNAs silence htt mRNA with a moderate reduction in potency compared to pan-HTT targeting siRNAs. HeLa cells were treated with siRNAs via passive uptake for 72 hours. Huntingtin mRNA levels were measured with Quantigene 2.0 assay, and normalized to *PPIB*.

“The SNP targeting siRNAs is two to four-fold less potent than the most active non-selective siRNAs in the native genomic context

Extensive screening across the whole gene can identify highly potent non-selective siRNAs but the sequence space available for identifying SNP targeting compounds is limited. By screening hundreds of siRNAs, we have previously identified a highly active non-selective siRNA^{HTT} targeting HTT⁵⁰. HeLa cells express endogenous HTT and have the A isoform at the rs362273 SNP site. Therefore, we can compare the potencies of SNP- and non-selective siRNAs against the endogenous target. We evaluated the efficacy of SNP4, SNP6, and siRNA^{HTT} in HeLa cells at seven concentrations (Supplementary Figure 2). In passive uptake, the corresponding IC₅₀ values for SNP4, SNP6, and siRNA^{HTT} were 302nM, 149nM, 73nM, indicating that limiting the targeted sequence space by targeting a SNP resulted in at least a two-fold reduction in potency.”

3. Do one of the fully humanized HTT mouse models contain the appropriate SNP pairs to evaluate selectivity in vivo? Although the siRNA does recognize the mouse htt transcript, there are significant sequence differences between human and mouse HTT genes that could have non-obvious effects on the potency of the ASOs.

This is a great suggestion, and an experiment to do in the future. At this point, these models are unfortunately not widely available. However, we have now included data from patient-derived NSCs and neurons differentiated from patient-derived iPSCs (data is included in response to comment #1, above), which shows that the siRNA recognizes and is specific in a fully human context.

4. Suggest also comparing against the non-allele selective siRNA published in the Alterman et al. manuscript in the mouse model.

The modified version of the manuscript contains new data directly comparing the optimized siRNA^{HTT} (identified by Alterman et al) in HeLa cells (see comment #2, above) and in patient NSCs (see comment #1, above). There is a measurable loss of efficacy (2-4 fold) with the SNP-targeting compounds. This does potentially translate to a need to use higher doses *in vivo* to achieve similar levels of silencing.

5. It is not possible to calculate fold selectivity based on a single dose. Suggest doing a dose response in the mouse model.

We have now modified the manuscript and included efficacy data at two doses levels, 225µg and 450 µg. 450 µg approaches the maximum deliverable dose due to solubility/viscosity limitations. We clearly see that greater target silencing (95% vs 85%) is observed at the 450µg dose. Importantly, at the highest dose where mutant protein is silenced below the level of detection, we observed no detectable impact on wild-type *Htt* expression, confirming selectivity.

6. The upregulation of the normal mouse htt transcript warrants further investigation/discussion. This does not appear to occur when the mutant htt RNA is reduced with RNase H oligonucleotides. Does this occur at the RNA level as well? Does it occur in human cells? Cortical neurons from the transgenic mice? This effect seems to be most prevalent in striatum where the authors also reported evidence of siRNA-induced astrocytosis. Is this related to a toxicity? Does it occur with a non-allele selective siRNA that only targets the human transgene?

We have further evaluated this upregulation and, while it is a potentially interesting phenomenon, we can now show that it does not occur in the fully human context (in patient-derived cells). Therefore, the finding is not relevant to this study. We have removed the discussion from the text.

7. For the mouse studies- is the listed dose the total dose administered or dose per ventricle?

The dose reported is the total dose. We have clarified the text accordingly. "We administered 225 µg (10 nmol, 5 nmol/ventricle) of divalent SNP6-11 into the lateral ventricles of BACHD mice."

8. Did the authors perform a histological analysis of brain tissue to see if there are any notable histological changes in the striatum that correlate with increased GFAP? Was AIF1 measured?

There are no gross changes in brain histology. The impact of the chemistry has been systematically evaluated elsewhere (<https://www.nature.com/articles/s41587-019-0205-0>).

To address the reviewer comment we evaluated the levels of Iba1 levels in the cortex and striatum and show no changes in expression. The new data (referenced below) is included as a new Supplementary Figure 8.

Supplementary Figure 8. SNP6-11 brain administration does not elicit microglial activation as indicated by lack of IBA1 elevation. BAC-HD mice were injected with 225 μ g of SNP6-11, NTC siRNAs and PBS. Levels of IBA1 expression in striatum and medial cortex, evaluated by automated western blot and normalized to GAPDH loading control. N=5-6, one-way ANOVA with Tukey multiple comparison correction.

“Selective mutant *htt* silencing did not change the levels of the neuron specific protein Dopamine- and cAMP-regulated phosphoprotein (DARPP32; Supplementary Figure 7a). Glial fibrillary acid protein (GFAP), a marker of astrocytes, was also unchanged, except in the striatum (Supplementary Figure 7b). Levels of ionized calcium binding adaptor molecule (Iba1), which localizes to microglia, were not different between treatment groups (Supplementary Figure 8), indicating that exposure to the divalent siRNA did not elicit significant microglial activation at the times tested.”

9. Should reference the original BAC97 paper in methods.

We apologize. We have added this reference to the methods.

10. Reference 18 and 45 appear to be duplicates.

This error has been corrected.

Reviewer #2 (Remarks to the Author):

The manuscript “Chemical engineering of siRNAs for allele-specific gene silencing in vivo in CNS” by Conroy et al. makes use of mismatch mediated selectivity of siRNAs for Huntington disease model. The concepts and rationale have been well-thought out and has been done in a logical manner. Before this article is accepted for publication, the following questions need to be addressed:

1. The authors state “chemical and thermodynamic mediated optimization” but there is no experiment or data to explore chemical modification and mismatch-related thermodynamic binding affinity. The authors need to measure the thermodynamic melting temperatures for the chemically modified RNA strands against their RNA target sequences. Such a method has been shown in literature for this gene before and it is a doable experiment. This will explain

- a. why the selectivity changes occur in going from one sequence to the other,*
- b. why double mismatch works better than a single mismatch and*
- c. why certain mismatch base pairs (e.g., C:U vs G:U) work better than others*
- d. why 6-11 double mismatch works better than other mismatches*

To address the reviewer’s comment, we have performed additional experiments and evaluated the impact of a single mismatch on the guide strand to target overall thermodynamics. Using a thermo stability assay we evaluated the impact of the single mismatch in position 6 on the T_m of the guide-target RNA complex in the context of a 19-mer target and local 13-mer interactions. As expected, for the fully modified strand, the relative contribution to the overall T_m is minor (4-5°C, (85.4°C to 81.4°C reduction)), which doesn’t affect affinity in a biologically significant way. Thus, the mismatch induced discrimination is not driven by changes in the overall affinity, but by changes in the local target recognition architecture. This is now included as Supplementary Figure 2, which is reproduced below along with the additional text from the results section.

Supplementary Figure 3: A single mismatch in the seed region of the siRNA guide strand does not have a significant impact on duplex stability. Guide strands for SNP6-11 were hybridized to 19 nucleotide and 13 nucleotide RNA strands, with or without a mismatch in the seed, and melting temperature was measured by thermo stability assay. **(a,d)** Sequence and structure of the 6-11 guide strand and a complementary 19 nucleotide RNA or 13 nucleotide RNA. **(b,e)** T_m curves for SNP6-11 hybridized to a 19mer complementary RNA or 13mer complementary RNA strand, with a single mismatch or full matched sequence. **(c,f)** Graph comparing melting temperatures of the full match sequence with a single mismatch included, exhibiting a minimal change in melting temperature, whether hybridized to a 13nt or 19nt complementary RNA strand.

“The level of siRNA accumulation *in vivo* varies significantly between different brain regions and over time. Therefore, a single mismatch at the optimal position (SNP6) likely provides insufficient discrimination to support allele-specific silencing *in vivo*. In the context of a fully modified guide strand, a single mismatch introduces only a small thermodynamic disturbance, and indeed in certain positions, a single mismatch

between the siRNA and the target mRNA can enhance efficacy⁵¹. We measured the impact of the mismatch in position six on the T_m of the guide strand/RNA substrate (Supplementary Figure 3). In the context of the full-length guide, the impact of the mismatch on stability was 4°C (85.4°C to 81.4°C), which is not biologically significant. In the context of a 13-mer substrate (comprising the RISC core interactions), the impact of the mismatch was greater (5.1°C—68.5°C to 73.6°C). However, in both cases, the affinity of the modified guide strand for the target is high. The likely explanation for the observed discrimination is a disruption of the local architecture of the siRNA-target duplex, which interferes with the ability of RISC to form the active conformation and disrupts scanning and recognition of the target by the seed region (positions 2-8)^{52–54}.

2. The oligonucleotide synthesis details should include whether the duplexes have regular phosphates or E-vinyl phosphonates at the 5'-end, where the PS modifications are and if they have a lipophile. A simple Table showing duplexes with chemical modifications will address this problem. Such confusion arises because in p-4 it is stated "All terminal backbones were modified with phosphorothioates (PS) (Figure 1a). - When cholesterol-conjugated, this type of compound is readily internalized by many cell types in vitro. The same chemical scaffold can later be re-synthesized in a divalent configuration for delivery to the CNS, allowing seamless transition between in vitro and in vivo experimental systems." whereas Figure 1 does not show full P=S and does not show cholesterol conjugate. Also, discuss the biodistribution differences between PS backbone RNA vs cholesterol conjugates.

Supplemental Table 1 now shows all sequences with chemical modifications used in the paper. The table also includes, for each modification pattern and sequence, the figures in which it was used. In addition, we included a graphical representation of the oligonucleotide structure/chemical modification patterns used *in vitro* and *in vivo* within figures 1 (reproduced below) and 5 (in response to the reviewer's comment #4).

Figure 1. The primary screen for optimal siRNA sequence yields potent compounds with moderate discriminating power. (a) siRNA structure and chemical modification pattern used in screening (b) The primary screen identifies the most favorable positions of the SNP enabling single nucleotide discrimination for targeting of SNP site rs362273 (highlighted in red). Compounds were tested using a dual-luciferase reporter assay system in HeLa cells. The (psiCheck) reporter plasmids contain a 40mer region of huntingtin, including the target (A) SNP (black), and non-target (G) isoform (red). Cells were treated for 72 hours at 1.5 μ M of siRNA. A panel of siRNA sequences were synthesized in a cholesterol-conjugated scaffold with phosphorothioate and alternating 2'-F and 2'-OMe backbone modifications. By walking the siRNA sequence around SNP site rs362273, we find multiple compounds with varying degrees of efficacy and discrimination. (c) 7-point dose-response shows that siRNAs with the SNP site in positions 4 (SNP4) and 6 (SNP6) generate 20-fold and 7-fold allelic discrimination, respectively, with a high degree of efficacy.

“We designed a panel of twelve chemically stabilized siRNAs, overlapping SNP rs362273 of the huntingtin mRNA (Figure 1). Supplementary Table 1 shows the sequences and chemical modification patterns of all compounds used in the study. Alternating 2'-O-methyl (2'-OMe) and 2'-deoxy-2'-fluoro (2'-F) replaced the riboses, and terminal backbones were modified with phosphorothioates (PS); the sense strand was conjugated to cholesterol (Figure 1a). We refer to these siRNAs as fully chemically modified.”

3. The in vitro experiments have been done with monovalent siRNAs whereas the in vivo testing is done with divalent siRNAs: is there a correlation of monovalent to divalent siRNAs in vitro? Why the divalent molecules were not tried for in vitro experiments?

As the chemical structure and modification patterns are not changed, the *in vitro* efficacy in lipid-mediated delivery of monovalent and divalent siRNAs is identical (<https://www.nature.com/articles/s41587-019-0205-0>. Supplementary Figure 2)

4. Again, clearly show the structure of the divalent siRNAs used for the in vivo experiment in Figure 5.

- a. Does it have a phosphate or vinylphosphonate?*
- b. Does it have cholesterol molecules or just P=S linkages?*

Figure 5 has been modified to include the di-valent structure with modification patterns shown.

Figure 5. SNP6-11 selectively silences mutant HTT in an HD mouse model. (a) BAC97 HD mice have a high copy number of transgenic mutant human huntingtin (*Htt*) with a polyQ expansion of 97 repeats, and two normal copies of wild-type (WT) mouse *Htt*. Both the human BAC97 huntingtin and the WT mouse *Htt* have homology at the SNP6-11 target site, except for a heterozygosity at SNP site rs362273, mimicking the WT/mutant *Htt* SNP heterozygosity in

patients. **(b)** for *in vivo* delivery, the SNP 6-11 compound was synthesized in a di-valent configuration as depicted **(c)** When treated with 225µg (10nmols) of siRNA (SNP6-11) in a divalent scaffold, allele-specific silencing of mutant HTT protein in mouse brain is achieved one month after bilateral injection into the lateral ventricles (ICV injection). (d) Across brain regions, WT mouse HTT is preserved with no significant silencing detected (except in striatum), while transgenic mutant HTT is lowered significantly, averaging 70% silencing across brain regions (p-value <0.00001). Protein levels were measured via WES Protein Simple. A two-way ANOVA with multiple comparisons was performed for all protein analysis, comparing treatment groups to the PBS control for each brain region.

c. Also explain, Why a control divalent siRNA is not used in this in vivo experiment

Non-Targeting Control (NTC) is a mandatory control. We used a divalent NTC control with a chemical configuration identical to SNP6-11. The text is modified to make this clear.

“We administered 225 µg (10 nmol, 5 nmol/ventricle) of divalent SNP6-11 into the lateral ventricles of BACHD mice. PBS and a chemically matched divalent NTC siRNA were used as negative controls.”

5. Explain the 2'-OMe vs 2'-F based activity differences based on Ago2 loading, thermodynamics, and relative metabolic stability of the siRNAs used. Metabolic stability of the OMe vs F construct should be evaluated in mouse brain tissues.

The relative stability 2'-OMe vs 2'-F has previously been studied, with 2'-OMe providing a higher level of stability compared to 2'-F. The study by Foster et al (<https://pubmed.ncbi.nlm.nih.gov/29456020/>) is an example which examines the relative contributions of different modification patterns on *in vivo* activity and duration of effect. We have modified the text of the paper to make these points clearer (see below). The relationship between accumulation and Ago2 potency is complex in nature and is the subject of detailed investigation in the lab, which is outside the scope of this paper.

“While both 2'-F and 2'-OMe stabilize the siRNA, the impact of 2'-OMe modification on stability is significantly more pronounced⁵⁵. Thus, increases in 2'-OMe content improve siRNA stability and enhance duration of effect *in vivo*^{16,56}, but extensive 2'-OMe modifications can position and sequence-specifically decrease activity⁵⁷. Therefore, most siRNAs designed for *in vivo* and clinical applications use sequence-optimized patterns of 2'-OMe and 2'-F modifications⁴. Successful development of SNP targeting siRNA requires an understanding of how the chemical modification pattern around the mismatch affects efficacy and discrimination. 2'-OMe and 2'-F affect guide strand/target interactions differently. 2'-F strengthens target interaction substantially more than does 2'-OMe^{58,59}. An increase in 2'-F content, particularly around the mismatch site, is expected to increase local affinity of the siRNA for its target, reducing the impact of a mismatch.

Meanwhile, the bulkier 2'-OMe might increase discrimination by reducing silencing of the non-targeted isoform.”

6. Explain why ICV is used for in vivo experiments? This is not a viable method for potential therapeutic applications. Has this experiment been tried by intrathecal (IT) dosing? Does the divalent molecule reach different parts of brain by IT administration? Does it show the same selectivity?

Absolutely, clinically, IT is a preferred route of administration. Moreover, in large animals, the impact of placement of the CSF infusion is limited. Distribution is more affected by volume. In contrast, in rodents, the placement of the CSF infusion has a profound impact on distribution, both for the ASO and siRNAs, with IT delivering more to the spinal cord and limited distribution to deep brain structures. Taking into account the technical challenges associated with reproducible IT injections, ICV is widely used for the administration of ASOs and siRNAs in rodents models, while IT is the preferred route of administration in large animals and for clinical use.

We have a paper in JCV insight <https://pubmed.ncbi.nlm.nih.gov/34935646/> systematically evaluating the species-specific differences in the impact of route of administration on divalent siRNA distribution and efficacy.

7. References 36 and 37 are the same.

Fixed

8. References 18 and 45 are the same.

Fixed

Reviewer #3 (Remarks to the Author):

This manuscript by Khvorova and colleagues explores possible variations of htt-targeting siRNA design that can render these SNP-based, allele-targeted siRNAs more allele-specific. This is not only an important elucidation of modifications allowing selective targeting mutant versus wild-type htt alleles, but also an excellent contribution to the design of SNP-targeted siRNAs in general. The former consideration is important and timely given the high interest in clinical application of htt-targeting gene therapies for Huntington's disease. In addition, the latter renders the article of high general interest with regard to the rationale design of SNP-targeted gene therapies in general.

The manuscript is concise and well-written. The conclusions are solid overall and these will have important gene therapy applications.

I was also pleased to read the authors' careful consideration of the upregulation of the endogenous, WT allele in parallel with the reduction in mutant HTT protein. Overall, this is an excellent study that will contribute meaningfully to the previous literature in the fields of Huntington's disease therapies, and design of siRNA-based therapeutic agents in general.

The relevance of these findings to current gene therapy strategies for counteracting the neuropathogenesis of mutant HTT warrants particularly rapid publication.

As a very minor revision, I would like to see the authors include further reflection on the possible significance of the non-sequence-specific effect on the upregulation of GFAP in an important Huntington's disease-affected brain region (striatum). At the authors' discretion, I would also welcome further experimental analyses to establish whether or not this might have been accompanied by other signs of gliosis or inflammation.

To address this comment, we have evaluated the levels IBA1 (a marker of microglial activation) in the mouse brains. New supplementary Figure 8 (reproduced below) shows the data. We have now optimized the formulation/chemical architecture of divalent compounds to eliminate temporary GFAP activation observed in early studies.

Supplementary Figure 8. SNP6-11 brain administration does not elicit microglial activation as indicated by lack of IBA1 elevation. BAC-HD mice were injected with 225 µg of SNP6-11, NTC siRNAs and PBS. Levels of IBA1 expression in striatum and medial cortex, evaluated by automated western blot and normalized to GAPDH loading control. N=5-6, one-way ANOVA with Tukey multiple comparison correction.

Overall, this is an excellent study that will contribute meaningfully to the previous literature in the fields of Huntington's disease therapies, and design of siRNA-based therapeutic agents in

general.

The relevance of these findings to current gene therapy strategies for counteracting the neuropathogenesis of mutant HTT warrants particularly rapid publication.

REVIEWERS' COMMENTS

Reviewer #1 (Remarks to the Author):

The authors have addressed my scientific concerns. However, I still remain unconvinced that this is the right journal for this publication.

Reviewer #2 (Remarks to the Author):

The reviewer comments and questions have been satisfactorily addressed.

Allele-specific gene silencing in vivo in CNS is a much needed research progress and looking forward to next steps by authors for well-planned clinical entry and subsequent success for needy patients.

The manuscript is recommended for publication.

Reviewer #3 (Remarks to the Author):

The recommendations of all three reviewers have been fully addressed, including relevant new data.

This paper should be published as soon as possible to benefit the field.